# The dark side of the forces: assessing non-conservative force models for atomistic machine learning

**Filippo Bigi** [1]   **Marcel F. Langer** [1]   **Michele Ceriotti** [1]

## Abstract

The use of machine learning to estimate the energy of a group of atoms, and the forces that drive them to more stable configurations, has revolutionized the fields of computational chemistry and materials discovery. In this domain, rigorous enforcement of symmetry and conservation laws has traditionally been considered essential. For this reason, interatomic forces are usually computed as the derivatives of the potential energy, ensuring energy conservation. Several recent works have questioned this physically constrained approach, suggesting that directly predicting the forces yields a better trade-off between accuracy and computational efficiency, and that energy conservation can be learned during training. This work investigates the applicability of such non-conservative models in microscopic simulations. We identify and demonstrate several fundamental issues, from ill-defined convergence of geometry optimization to instability in various types of molecular dynamics. Given the difficulty in monitoring and correcting the lack of energy conservation, direct forces should be used with great care. We show that the best approach to exploit the acceleration they afford is to use them in conjunction with conservative forces. A model can be pre-trained efficiently on direct forces, then fine-tuned using backpropagation. At evaluation time, both force types can be used together to avoid unphysical effects while still benefitting almost entirely from the computational efficiency of direct forces.

[1]Laboratory of Computational Science and Modeling, IMX, École Polytechnique Fédérale de Lausanne, 1015 Lausanne, Switzerland. Correspondence to: Michele Ceriotti <michele.ceriotti@epfl.ch>.

*Proceedings of the $42^{nd}$ International Conference on Machine Learning*, Vancouver, Canada. PMLR 267, 2025. Copyright 2025 by the author(s).

## 1. Introduction

Interatomic potentials model the microscopic interactions between atoms, and determine – directly or by modulating the response to thermal excitations – the stability and reactivity of molecules and materials. Over many decades, interatomic potentials have been used in Monte Carlo (Metropolis et al., 1953) and molecular dynamics (MD) (Andersen, 1980) simulations, geometry optimization, and other techniques, allowing a mechanistic study of the atomic-scale behavior and properties of molecules, materials, and biological systems (Allen & Tildesley, 2017; Tuckerman, 2008). While traditional interatomic potentials are based on simple physically inspired functional forms, the last decade has seen machine-learned interatomic potentials (MLIPs) obtain remarkable accuracies at a high level of computational efficiency, most often by learning reference energies and forces from much slower first-principle quantum mechanical calculations (Behler, 2021; Unke et al., 2021).

At first, such machine-learned interatomic potentials were trained on quantum mechanical data specific to the chemical system of interest. Over the last years, more diverse datasets have emerged, consisting of up to billions of targets, and spanning much of the periodic table (Chanussot et al., 2021; Tran et al., 2023; Barroso-Luque et al., 2024; Schmidt et al., 2024). This has not only led to an increase in the complexity of the models, going from linear or kernel regression to deep graph neural networks, but it has also pushed many recent models to abandon some of the underlying physical symmetries of interatomic potentials in favor of simpler – and potentially more scalable and efficient – architectures (Pozdnyakov & Ceriotti, 2023; Neumann et al., 2024; Qu & Krishnapriyan, 2024). In such models, symmetries are learned from data, aided by data augmentation.

Some recent models for interatomic potentials also disregard the property of energy conservation (Gasteiger et al., 2021; Neumann et al., 2024; Liao et al., 2023). While conservative models calculate interatomic forces as the derivatives of a total energy with respect to the positions of the atoms, non-conservative models predict them directly, therefore breaking this constraint. Although this practice can lead to more computationally efficient neural networks, the use of such models in practical atomistic simulations has not yet

been studied in detail.

This work compares conservative and non-conservative machine-learned interatomic potentials. Following a brief review of the most common types of potentials and their applications, we discuss the theoretical implications of using non-conservative forces to drive atomistic simulations, highlighting potential shortcomings. We then demonstrate the impact of these conceptual problems in several case studies, highlighting the pitfalls of using non-conservative interatomic forces in production, and demonstrating how, contrary to the case of geometric symmetry breaking, it is difficult to monitor and correct the impact of non-conservative models. Instead, it might be preferable to supplement a conservative model with direct force predictions, using them to accelerate simulations that have well-defined, energy-conserving forces as the ground truth.

## 2. Background and related work

### 2.1. Interatomic potentials and atomistic simulations

An interatomic potential $V$ describes the potential energy between the $N$ atoms of a molecule or material as a function of their positions $\{\mathbf{r}_i\}_{i=1}^N$ and chemical nature $\{a_i\}_{i=1}^N$:

$$V(\{\mathbf{r}_i, a_i\}_{i=1}^N). \tag{1}$$

While some applications, such as Monte Carlo simulations or energy difference calculations (for transition state or defect energies, material stability, etc.) need only the values of $V$, the most popular uses of interatomic potentials require the evaluation of interatomic forces, defined as the negative derivatives of $V$ with respect to the atomic positions:

$$\mathbf{f}_j = -\partial V(\{\mathbf{r}_i, a_i\}_{i=1}^N) \,/\, \partial \mathbf{r}_j. \tag{2}$$

The most notable applications which make use of interatomic forces are:

**Geometry optimization.** This technique consists of finding one or more minima of the potential energy surface $V$ to identify the preferred structure of a microscopic system. Both local and global optimization algorithms are employed, and the vast majority use the forces $\mathbf{f}_j$ both during optimization and as a stopping criterion. Similar methods are used to identify saddle points of $V$, which are associated with the energy barriers that determine the rate of chemical reactions (Henkelman et al., 2000).

**Molecular dynamics.** MD aims at simulating the behavior of a microscopic system by solving its classical equations of motion numerically. Using a time step $\Delta t$, the simplest forms of molecular dynamics propagate a discretized version of Hamilton's equations (Verlet, 1967). Over more than 50 years, many variants of molecular dynamics have

emerged, with various goals (sampling different thermodynamic ensembles (Andersen, 1980), improving sampling efficiency, observing rare events (Laio & Parrinello, 2002), or accounting for nuclear quantum effects (Chandler & Wolynes, 1981), etc.). In this work, we focus on some of the simplest variants:

*MD in the NVE ensemble.* The number of atoms $N$, volume $V$ and total energy $E$ (kinetic plus potential) are kept constant, describing the behavior of an isolated system. Usually, this is accomplished by using the velocity Verlet integrator (Verlet, 1967) with a short time step $\Delta t$.

*MD in the NVT ensemble.* Here, the goal is to simulate a system at constant temperature $T$. This is achieved by modifying the dynamics with so-called thermostats. *Local* thermostats apply velocity corrections on each atom independently (Bussi & Parrinello, 2007). While they correctly sample the desired constant-temperature ensemble and can be used to evaluate static, equilibrium properties, this comes at the cost of disrupting the dynamics of the system. *Global* thermostats (Bussi et al., 2007) address this shortcoming by modifying the atomic velocities in a concerted manner, making it possible to investigate time-dependent properties (Bussi & Parrinello, 2008).

Many other applications of interatomic potentials (such as lattice dynamics, phonon calculations, etc.) require the calculation of forces and their higher-order derivatives. However, for simplicity, this work focuses on the most common use cases, discussed above.

### 2.2. Physical symmetries

Interatomic potentials obey a number of physical symmetries and constraints.

**E(3)-invariance.** Interatomic potentials are invariant under the transformations of the Euclidean group in three dimensions E(3), including translations, rotations, and reflections. Mathematically, given a group element $g \in E(3)$ that acts on all positions $\mathbf{r}$,

$$V(\{g \cdot \mathbf{r}_i, a_i\}_{i=1}^N) = V(\{\mathbf{r}_i, a_i\}_{i=1}^N). \tag{3}$$

**Permutation invariance.** Interatomic potentials are invariant with respect to permutations of atom indices, i.e.,

$$V(.., \mathbf{r}_i, a_i, .., \mathbf{r}_j, a_j, ..) = V(.., \mathbf{r}_j, a_j, .., \mathbf{r}_i, a_i, ..). \tag{4}$$

**Energy conservation.** Interatomic forces are conservative, i.e., their mechanical work $W$ over a closed loop is zero:

$$W = \oint \sum_{j=1}^N \mathbf{f}_j \cdot \mathrm{d}\mathbf{r}_j = 0. \tag{5}$$

This is a direct consequence of (2). The term *conservative* alludes to the fact that an isolated physical system in which only conservative forces act obeys the principle of conservation of mechanical energy.

## 2.3. Interatomic potentials via graph neural networks

For many years, most machine learning interatomic potentials made use of feature-based models (Behler & Parrinello, 2007; Bartók et al., 2010) trained on energies and forces from quantum mechanical calculations. This combination of physically inspired descriptors (Musil et al., 2021; Langer et al., 2022) and classical machine learning methods (linear and kernel models, as well as shallow neural networks) delivers good accuracy and computational efficiency on small datasets, usually created for a specific application in chemistry and/or materials science.

However, recent efforts to build much larger and more comprehensive datasets (Deng et al., 2023; Schmidt et al., 2024; Eastman et al., 2023; Tran et al., 2023; Chanussot et al., 2021), have made it apparent that graph neural networks exhibit superior accuracy and scalability compared to earlier models (Batzner et al., 2022), as they respect the permutation symmetry of interatomic potentials while delivering rich and flexible representations of the atomic structures. For a review of graph neural networks, we redirect the interested reader to Zhou et al. (2020), while their applications to interatomic potentials is detailed in Duval et al. (2023).

## 2.4. Breaking physical constraints

Several recent architectures do not enforce all physical constraints listed in 2.2, aiming to increase expressivity, ability to scale to large datasets, and computational efficiency. Efficient inference is crucial to enable the length and time scales required for practical molecular dynamics simulations, as forces must be evaluated at every time step.

**Rotationally unconstrained models.** In the case of rotational symmetry this approach, combined with training-time rotational augmentation, has been shown to yield accurate and efficient models (Pozdnyakov & Ceriotti, 2023; Neumann et al., 2024; Qu & Krishnapriyan, 2024) with negligible, or easily controllable, impact on physical observables (Langer et al., 2024).

**Non-conservative models.** The conservative property for forces in Eq. (5) is satisfied if and only if there exists a function (usually named the potential energy) of which the forces are the spatial derivatives, see Eq. (2). Therefore, a machine learning model that calculates interatomic forces as derivatives of the potential energy according to Eq. (2) will be trivially conservative. The required derivatives, the gradient of the scalar $V$, can be obtained efficiently (i.e.,

with the same asymptotic computational cost as the energy prediction) with automatic differentiation (Griewank & Walther, 2008). Nevertheless, differentiation incurs a computational overhead, typically $2-3\times$ for inference and $3\times$ for training; the exact theoretical factors are discussed further in Appendix B. This overhead can be avoided by directly predicting forces during the forward pass. However, by removing the relationship between forces and energy defined in Eq. (2), such models do not enforce energy conservation. The possibility of performing a direct, non-conservative, evaluation of the forces was realized early (Li et al., 2015), but used only sparsely until recently, when it was applied to equivariant graph neural networks such as GemNet (Gasteiger et al., 2021) and several more recent universal machine learning interatomic potentials including ORB (Neumann et al., 2024) and Equiformer (Liao et al., 2023), which are trained on large datasets and promise broad applicability for chemical modeling. Some of these architectures also perform direct evaluation of stresses, which leads to the violation of conservation of enthalpy (see App. G.3).

The impact of lack of energy conservation for practical simulations and property prediction has recently been investigated in related works: Eissler et al. (2025) probe the limits of unconstrained architectures and find that lack of energy conservation becomes more pronounced for larger target systems. Loew et al. (2024) find that direct-force models perform poorly at the prediction of phonon properties, which require second derivatives of the potential energy surface. Fu et al. (2025) observe a better correlation between test set error and downstream predictive performance, for instance for phonons, for models that can perform stable MD simulations, i.e., that are able to conserve energy.

## 2.5. Use of forces for training

Machine learning models of $V$ are typically trained jointly on energy and force labels. The relative weight of these labels in training, or in the extreme case, whether to train exclusively on energy or on forces, has been discussed extensively, both for interatomic potentials and diffusion models for atomistic systems (Chmiela et al., 2018; Christensen & von Lilienfeld, 2020; Wang et al., 2024; Ren et al., 2024).

One important consideration is that focusing on energies or forces emphasizes different components of a potential energy surface, with consequences that are not directly visible when inspecting training and validation accuracies. To a first approximation, the distortions observed in NVT MD simulations – a common strategy to build training sets – can be interpreted as a collection of quasi-harmonic oscillators of different frequency $\omega$, with high-frequency modes associated with short-range covalent bonding, and low-frequency ones associated with collective motions, which are often the most relevant for applications. The statistical mechanics of

a harmonic oscillator imply that $\langle V \rangle \propto 1$, while $\langle f^2 \rangle \propto \omega^2$ (see Appendix A). In other terms, the largest contributions to the forces from a thermally sampled dataset come from high-frequency modes that are hard to integrate and may lead to instabilities in the dynamics, while the contribution from the slow modes, which are hard to sample, but usually the most relevant, are under-emphasized. The potential energy provides a more balanced representation of the different molecular time and length scales.

The question of relative energy and force weights during training is related to, but distinct from, the question of learning energy and forces consistently, i.e., enforcing that the predicted forces are exactly the gradient of the predicted energy. Models targeting mean-field energies, as in coarse-graining (Wang et al., 2019) and centroid MD (Musil et al., 2022), usually train exclusively on estimates of the mean forces, but use a conservative formulation as these forces are defined as derivatives of a thermodynamic potential that is difficult to evaluate explicitly. Disregarding the connection with reference energies may also be beneficial when the two targets are subtly inconsistent, because of convergence issues, the use of heterogeneous calculation settings, or numerical techniques like Fermi smearing (Marzari et al., 1999), which can be required for metallic systems. This may contribute to the empirical observation that non-consistent training is preferable in datasets like OC20 (Chanussot et al., 2021) and OC22 (Tran et al., 2023) that contain a high fraction of metallic configurations.

## 3. Theory

Some of the consequences of using non-conservative models descend directly from their mathematical formulation and can be used to foresee their impact on typical applications.

### 3.1. Measuring non-conservative behavior

If a vector field $\mathbf{f}$ is the derivative of a smooth scalar function $V$, then its Jacobian $\mathbf{J}$ (the Hessian of $V$) contains the mixed second derivatives of $V$, and must therefore be symmetric:

$$J_{i\alpha,j\beta} = \frac{\partial \mathbf{f}_{i\alpha}}{\partial \mathbf{r}_{j\beta}} = \frac{\partial \mathbf{f}_{j\beta}}{\partial \mathbf{r}_{i\alpha}} = J_{j\beta,i\alpha} \,. \tag{6}$$

In order to quantitatively capture the amount of non-conservation in a specific force prediction from a trained model, it is then possible to compare the Frobenius norm (or any other matrix norm) of the antisymmetric component of the Jacobian to that of the Jacobian itself:

$$\lambda = \frac{||\mathbf{J}_{\mathrm{anti}}||_{\mathrm{F}}}{||\mathbf{J}||_{\mathrm{F}}} \,, \tag{7}$$

where $\mathbf{J}_{\mathrm{anti}} = (\mathbf{J} - \mathbf{J}^\top)/2$. $\lambda$ then defines a metric going from 0 for conservative forces to 1 for forces that have

no conservative component. $\lambda$ can also be computed only for entries associated with an atom $i$, or a pair of atoms $ij$, providing a finer-grained assessment of the violation of Eq. 6, as seen in Figure 1.

This local symmetry breaking can also be measured by integrating the work done by the forces over a closed loop, that is bound to be zero (within integration error) for a conservative force field (Eq. 5). An explicit test of non-conservative behavior can also be implemented by monitoring the total energy in an NVE MD simulation, or equivalently a conserved quantity that keeps track of the heat flux associated with the thermostat (Bussi & Parrinello, 2007) in an NVT simulation. To find the power $P$ (energy per unit time) injected by the non-conservative forces during a molecular dynamics trajectory, it is sufficient to calculate the average rate of change in the conserved quantity $C$ during a section of the trajectory, $P = \Delta C / \Delta t$.

### 3.2. Side-effects of direct force prediction

Predicting forces as the derivatives of a translationally invariant potential ensures that the total net force acting on all atoms is zero, and that for a potential that is rotationally invariant the torque on an isolated molecule is zero. A direct force model – irrespective of whether it is E(3) invariant – does not have the same guarantees. These spurious effects are easy to remedy, by subtracting the total force and torque from each prediction. This technique is used by ORB (Neumann et al., 2024), and we also adopt it here.

There is another non-trivial consequence of using a direct-force prediction architecture. Whereas the potential energy is usually estimated as the *sum* of atomic contributions but is a global property, forces are atom-centered. When predicting them as derivatives, many atomic environments contribute to the force $\mathbf{f}_i$ on each atom. When predicting directly, only the $i$-atom centered environment contributes to $\mathbf{f}_i$. Hence, direct force models can be expected to be affected more directly by the geometric degeneracies of low-body-order atom-centered descriptors (Pozdnyakov et al., 2020) and do not benefit from the same extended effective interaction range as conservative forces (Artrith et al., 2011). We discuss these effects in Appendix C, presenting some empirical evidence that direct force models require a larger range to match the force accuracy of a comparable conservative model.

### 3.3. Effects on geometry optimization

In order to assess the stability of materials or molecules at low temperature, a common approximation is to search for minimum-energy configurations. This can be achieved by minimizing the potential energy $V$ as a function of the atomic positions – with most of the widely used algorithms relying (in some cases exclusively) on the value of the gradi-

ent. The lack of a consistent potential energy is problematic for most optimization schemes: those which require an explicit evaluation of the objective function (e.g., to perform line searches) cannot be used; those that just "follow the forces", relying on the vanishing magnitude of the force as a stopping criterion, can fail because non-conservative forces can keep driving indefinitely in the same direction, e.g., following closed loops with negative total work.

### 3.4. Effects on molecular dynamics

It is not uncommon to observe violations of energy conservation in MD simulations, since finite-timestep integrators violate the exact correspondence between the change in kinetic and potential energy along a trajectory. This is usually accepted, because (1) in well-designed simulations this leads to small fluctuations, and not to a run-away effect; (2) the notion of a *shadow Hamiltonian* (Hairer et al., 2006) ensures that simulations reach a steady state that is "statistically close" to that generated by an exact integrator; (3) thermostatting techniques can relatively easily control small integration errors, so that structural and dynamical observables are not affected significantly (Morrone et al., 2011). The fact that no underlying Hamiltonian can be defined for the dynamics generated by a non-conservative force field suggests that, in this case, artifacts might be more pronounced and harder to correct. For example, the symplectic nature of Hamiltonian dynamics is no longer valid (this is easy to see, for example, from Eq. 5.2, Chapter 6 of Hairer et al. (2006)), and the theorem of equipartition of energy (whose proof is also based on the existence of a Hamiltonian (Pathria, 2017)) does not apply. As we shall see, this is what we observe empirically.

### 3.5. Learning conservative behavior

The standard approach to making symmetries more easily learnable from data is to employ data augmentation at training time: A random element of the underlying symmetry group, for instance rotations, is selected and applied for every sample or mini-batch. This approach has been successfully employed in the domain of MLIPs, as well as in a range of other applications including computer vision (Quiroga et al., 2020). However, it is only applicable to explicit geometric symmetries, and therefore not to energy conservation, which is not a symmetry with respect to inputs, but rather a symmetry of derivatives, as discussed in Section 3.1. Nevertheless, we briefly consider different schemes to promote energy conservation during training.

One approach is to include the measure of Jacobian symmetry $\lambda$, Equation (7), as a term in the loss function. This faces severe practical issues: With automatic differentiation, computing $\mathbf{J}$ requires multiple ($3N$ in the absence of sparsity or stochastic approximations) evaluations of the potential.

An alternative approach is to train both a conservative and non-conservative predictor and adding a force-matching loss term, or simply training both on the same forces labels. We will discuss this approach in further detail in Section 4.8. It is important to note that, in both strategies, energy conservation can be trained both on labeled and unlabeled data.

## 4. Results

Having introduced the subject of non-conservative force fields and discussed the potential pitfalls that might be incurred when using them in practice, we will now examine their effect on a range of applications, using liquid water as the main, paradigmatic example.

### 4.1. The models

In order to substantiate our empirical observations, we perform our experiments on multiple models. Our main examples rely on the rotationally unconstrained PET architecture (Pozdnyakov & Ceriotti, 2023), trained from scratch on the bulk water dataset of Ref. 22 using both a conservative (PET) and non-conservative (PET-NC) architecture. Additionally, we train "PET-M" to predict both conservative and non-conservative forces. To assess the implications of a direct prediction of forces in the most favorable possible context, we primarily use the best-performing, custom-trained models. We also show some results for the non-conservative ORB-v2 model (Neumann et al., 2024), which is currently state-of-the-art for several materials prediction benchmarks. Even though ORB is not trained on this specific dataset, and is therefore at a clear disadvantage, it provides an indication of the relevance of the issues we discuss. In the appendices, we also discuss several other architectures, including a "legacy" SOAP-BPNN architecture, as well as pre-trained foundation models, MACE-MP-0 (Batatia et al., 2023), SevenNet (Park et al., 2024), and EquiformerV2 (Liao et al., 2023), which we apply to a few diverse materials in Appendix G. A table describing all employed models, along with full details on the different architectures, can be found in Appendix D.

### 4.2. Accuracy

In terms of sheer accuracy, see Table 1, our tests confirm that forces provide very useful information to train an interatomic potential, in particular for a dataset containing relatively large configurations. Using forces in the training of a conservative model dramatically improves the accuracy of energy predictions with just a minor degradation in the accuracy for $\mathbf{f}$ with respect to a model trained only on forces. A non-conservative force model exhibits about 30% higher force error than a conservative architecture, and including a separate energy head leads to lower error than a model trained just on $V$ – indicating that the sharing of

weights within the architecture is beneficial. We also show results for the PET-M hybrid architecture, which makes both conservative and non-conservative force predictions. Its accuracy is less than 10% worse than the best models for either architectures. As we will discuss in Section 4.8, this is an excellent way to exploit non-conservative forces in simulations. Before doing so, however, we will assess the behavior of purely non-conservative models.

### 4.3. Non-conservative behavior

The asymmetry of the Jacobian is the most direct, point-wise measure of non-conservative behavior. Different non-conservative models show widely different values of $\lambda$ – 0.015 for ORB, 0.017 for Equiformer, 0.032 for SOAP-BPNN-NC and 0.004 for PET-NC, computed on a few water structures from the test set of Cheng et al. (2019). As we shall see, the magnitude of $\lambda$ correlates qualitatively with the stability of the models in simulations. The symmetry of $\mathbf{J}$ applies separately to each pair of atoms, and so it is possible to extract further insights by computing the norm of the antisymmetric part of each block $\mathbf{J}_{ij}$ resolved for different atomic pairs and plotted as a function of the interatomic distance (Figure 1). The asymmetry is also present for "on-site" blocks, i.e., swapping only the Cartesian coordinates used in the derivatives for a given atom; the relative magnitude of $\mathbf{J}_{\text{asym}}$ is small compared to the magnitude of the Jacobian; the asymmetric component between atoms $i$ and $j$ decays with the interatomic distance more slowly than the absolute magnitude of $\mathbf{J}$ (which has been used as a measure of the interactions between pairs of atoms (Herbold & Behler, 2022)), and in the intermediate regime around 6Å it becomes comparable in size – so that the pair-resolved Jacobian asymmetry $\lambda_{ij}$ approaches 1 for large interatomic distances. This latter observation has important implications when applying these models in simulations, as the impact of non-conservative behavior on different atomic-scale processes is not uniform, and it tends to be larger – in a relative sense – for collective processes involving long-range correlations.

*Table 1.* Test errors (energies in meV per atom, forces in meV/Å) of PET models trained on the bulk water dataset. In this table, and all subsequent tables, rows corresponding to non-conservative models are highlighted with a gray background.

| Arch. | Type | Training | MAE($V$) | MAE($\mathbf{f}$) |
|---|---|---|---|---|
| PET | – | $V$ | 4.7 | – |
| PET | C | $\mathbf{f}$ | – | 18.6 |
| PET | C | $V, \mathbf{f}$ | 0.55 | 19.4 |
| PET | NC | $\mathbf{f}$ | – | 24.3 |
| PET | NC | $V, \mathbf{f}$ | 1.42 | 24.8 |
| PET-M | C | $V, \mathbf{f}$ | 0.59 | 20.2 |
| | NC | | | 26.7 |

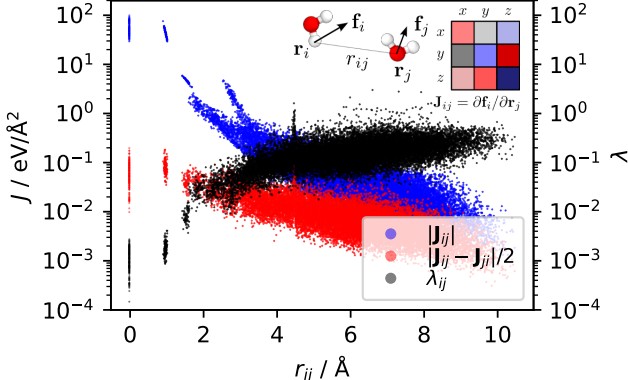

*Figure 1.* Comparison of the norm of each block of the Jacobian, $\mathbf{J}_{ij}$, and of its antisymmetric component, for different pairs of atoms, as a function of their distance $r_{ij}$, computed for a randomly selected bulk water structure from the test set.

Non-conservative behavior can also be demonstrated by computing the work along a closed path (see Appendix E). Given that the choice of the path is arbitrary, we think it is more relevant to quantify the practical implications of this issue in terms of an energy drift in molecular simulations.

### 4.4. Constant-energy molecular dynamics

Let us now consider the use of non-conservative MLIPs in the context of MD simulations. Also in this case, we focus our analysis on the best-performing models for liquid water, PET and PET-NC. A more thorough comparison, including several foundation models and a few homogeneous and heterogeneous material structures, is discussed in Appendix G, and consistently corroborates the observations we make here. Constant-pressure simulations, which, if performed with direct-stress models, break conservation of *enthalpy* and therefore lead to drift in the volume of the simulation, are shown in App. G.3 using the general-purpose PET-MAD potential (Mazitov et al., 2025).

Given that non-conservative models lack a well-defined conserved quantity by construction, we rely on indirect measurements of the sampled ensemble. We consider in particular the kinetic temperature $T = 2K/(3Nk_{\text{B}})$, where $N$ is the number of particles considered. This is just a rescaling of the kinetic energy $K$; its ensemble average should correspond to the target temperature for NVT trajectories (300 K in these tests), and to a value in its vicinity for NVE trajectories initialized from a thermally equilibrated configuration. As a sensitive indicator of the dynamical behavior of the system, we compute the Fourier transform of the velocity-velocity correlation function, $\hat{c}_{vv}(\omega)$. Its peaks are closely related to the density of vibrational modes and to infrared and Raman spectra, and its $\omega \to 0$ limit is proportional to the diffusion coefficient.

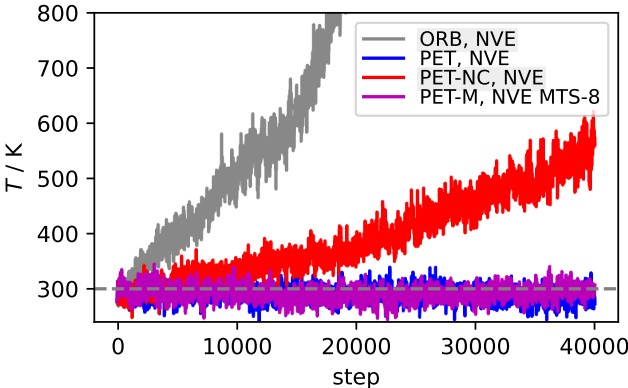

*Figure 2.* Time series for the kinetic temperature along a NVE MD trajectory, for ORB, the conservative and non-conservative PET models, and for a PET-M model using a multiple time-stepping (MTS) algorithm that evaluates conservative forces every 8 steps.

The failure of the non-conservative model in NVE dynamics is apparent in Figure 2. Whereas for a conservative potential the kinetic temperature fluctuates around the initial value, the spurious work associated with non-conservative forces leads to a large drift of $T$: To put it on a human scale, this unphysical drift corresponds to a rate of heating of about 7'000 billion degrees per second for the custom-trained PET-NC model, and another 10 times larger for the general-purpose ORB model. This spurious energy flow is a clear signature of non-conservative behavior, and makes direct-force models entirely useless for constant-energy simulations. This runaway increase of the kinetic energy can be mitigated – whenever an auxiliary model is available to evaluate the potential energy – by adjusting the velocities to enforce energy conservation artificially (see Appendix G.4). Similar to what we will discuss for constant-temperature simulations, the trajectories are still affected by large artefacts.

### 4.5. Equilibrium properties in the NVT ensemble

The very use of a finite time step in the integration of MD trajectories causes energy fluctuations, and it is not uncommon to use advanced simulation schemes that violate energy conservation (Kühne et al., 2007; Mazzola & Sorella, 2017; Morrone et al., 2011; Laio & Parrinello, 2002), for instance because they use approximations that yield forces contaminated by a stochastic noise. In these cases, using a thermostat can counterbalance the energy error, and obtain stable trajectories that yield configurations and equilibrium average properties close the correct NVT ensemble despite the drift of the conserved quantity that generalizes the total energy for constant-temperature simulations.

Judging by the average temperature $\langle T \rangle$ (Table 2), it is relatively easy to control the non-conservative behavior using a white-noise (WN) Langevin thermostat. However, strong couplings $\tau$ (the time scale over which the thermostat inter-

*Table 2.* Mean kinetic temperature excess $\langle \Delta T \rangle - \langle T \rangle - 300$ K, and atom-type resolved temperatures $\langle \Delta T_O \rangle$ and $\langle \Delta T_H \rangle$, for ORB, the conservative (C), non-conservative (NC) and multiple time-stepping (MTS) PET models (which evaluates conservative forces once every 8 steps). White-noise Langevin (WN) and stochastic velocity rescaling (SVR) thermostats are also compared.

| THRM. | TYPE | $\tau$/fs | $\langle T \rangle$/K | $\langle T_H \rangle$/K | $\langle T_O \rangle$/K |
|---|---|---|---|---|---|
| | | | ORB | | |
| WN | NC | 1000 | 51.0(6) | 60.4(5) | 33(1) |
| WN | NC | 100 | 4.2(2) | 5.9(3) | 0.9(3) |
| WN | NC | 10 | 0.4(1) | 0.6(1) | 0.1(1) |
| SVR | NC | 10 | 1.0(1) | 36.2(8) | -70(2) |
| | | | PET | | |
| WN | C | 100 | 0.1(2) | 0.0(2) | 0.3(3) |
| WN | NC | 1000 | 12.8(5) | 11.2(7) | 16.2(5) |
| WN | NC | 100 | 1.4(2) | 1.3(2) | 1.6(3) |
| WN | NC | 10 | 0.1(1) | 0.0(1) | 0.2(1) |
| SVR | C | 10 | 0.1(1) | -0.4(3) | 1.0(7) |
| SVR | NC | 10 | 0.3(1) | -4.4(3) | 9.9(6) |
| SVR | M-8 | 10 | 0.0(1) | -0.1(4) | 0.1(9) |

feres with atomic motion) are needed, as even at 100 fs the average temperature is significantly off the target value. We discuss how these deviations in the equilibrium temperature affect structural properties of water in Appendix G.5. The upshot is that for accurate models and strong thermostatting, the effects are small but noticeable. Furthermore, the strong Langevin thermostatting reduces the sampling efficiency, and so even with a respectable 1 ns trajectory, many simple structural averages are not fully converged.

### 4.6. Sampling efficiency and time-dependent properties

Aggressive Langevin dynamics is bound to dramatically change time-dependent properties, and in particular to reduce the diffusion coefficient – so that longer trajectories are needed to collect statistically independent atomic configurations. This slow-down is apparent when looking at the velocity correlation spectra (Figure 3). In the weak coupling regime (WN, $\tau = 1000$ fs) there is a (small) increase in diffusion coefficient relative to the reference, because of the unphysically higher temperature, while the high-frequency peaks corresponding to stretching and bending are only weakly perturbed. Stronger couplings alter the dynamics dramatically, and reduce the diffusion coefficient (and hence the efficiency in sampling slow, collective motion) by a factor of about 1.5 ($\tau = 100$ fs) and 5 ($\tau = 10$ fs), negating the inference speed-up of a non-conservative model – while making it impossible to accurately evaluate any time-dependent property.

A potential solution – applied often in similar cases, including to control the artifacts of non-invariant predictions of $V$ (Langer et al., 2024) – is to resort to a *global* thermostat

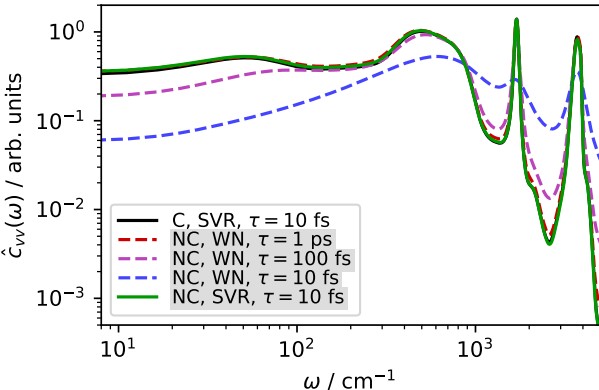

*Figure 3.* Velocity power spectrum $\hat{c}_{vv}(\omega)$, for different PET models and thermostat types: A conservative (C) and non-conservative (NC) model using white-noise Langevin (WN) and stochastic velocity rescaling (SVR) thermostats.

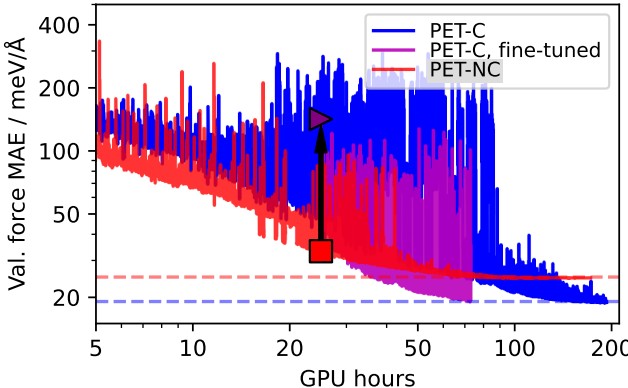

*Figure 4.* Training curves for PET-C and PET-NC models, and the conservative fine-tuning of a hybrid PET-M model initialized (epoch marked with arrow) from the potential energy head of the PET-NC model.

that only acts on the total kinetic energy rather than on individual particle momenta, achieving efficient temperature control without disrupting dynamics. We use the stochastic velocity rescaling (SVR) method (Bussi et al., 2007), which indeed brings the average temperature to within 1% of the target, without dramatically altering $\hat{c}_{vv}(\omega)$ even with a strong $\tau = 10$ fs coupling. However, a global thermostat cannot help when non-conservative terms act differently on the various degrees of freedom. This is evident in how the temperature of O and H atoms, computed separately, deviates by up to 10% from the target, which is reflected in loss of structure as measured by $g(r)$, and in an overestimation of the diffusion coefficient.

Some of the more sophisticated thermostats used to stabilize other types of MD approximations – such as carefully tuned generalized Langevin equations (Ceriotti et al., 2009; Morrone et al., 2011) – can also be used to enforce more aggressive temperature control with reduced dynamical disruption in conservative molecular dynamics, but they still modify the natural dynamical properties significantly, and they can fail catastrophically when used with non-conservative forces, as shown in Appendix G.6).

These experiments show clearly that while it is possible to mitigate the runaway temperature increase associated with the lack of energy conservation, doing so in a way that does not disrupt structural and/or dynamical observables is highly nontrivial or even impossible.

### 4.7. Geometry optimization

MD performed at very low temperature can be regarded as a form of geometry optimization. Our observations from MD suggest that sufficiently accurate non-conservative models should be able to reach reasonable, low-energy structures. We restrict ourselves to optimization algorithms based only

on gradients, to avoid the complication of using inconsistent energies, as discussed in Section 3.3, comparing the FIRE (Bitzek et al., 2006) algorithm – that is similar in spirit to zero-temperature MD – and LBFGS (Liu & Nocedal, 1989) – a quasi-Newton algorithm that uses an approximation of the Hessian to accelerate convergence. Comparing different models on the task of optimizing a water snapshot from a MD configuration (Figure 7) shows that inaccurate non-conservative models, such as SOAP-BPNN-NC, fail catastrophically at geometry optimization, while more accurate models, such as PET-NC and ORB, can reach a locally stable configuration, especially using FIRE. We note, however, that non-conservative models are less stable when used with a Hessian-based method, with large fluctuations in the residual force that make it hard to define a stopping criterion. On a practical level, non-conservative forces are bound to make geometry optimization more fragile, and to require careful choice of the minimization algorithm and its convergence parameters, as we observe in Appendix F when comparing different general-purpose models.

### 4.8. Non-conservative forces as accelerators

While we have observed that conservative MLIPs are better suited for practical simulations, we suggest that hybrid models, which additionally support direct, non-conservative, force predictions, can be used for faster inference and training. Such models can be obtained by training both force heads jointly, as demonstrated in the PET-M model, or, more efficiently, by first training a non-conservative model and then fine-tuning its energy head to yield conservative forces. As shown in Figure 4 and further discussed in Appendix H, conservative fine tuning leads to the accuracy and physical correctness of conservative models at highly reduced training time.

In simulations, one can then use the conservative forces of

such a hybrid model for validation, error monitoring and correction, and the direct forces for faster inference. A good example is to use multiple time-stepping (MTS) techniques (Tuckerman et al., 1992) for molecular dynamics, where the non-conservative forces are used to integrate the equations of motion, and the conservative forces are applied every $M$ steps as a correction. This reduces the theoretical overhead of a conservative trajectory from a factor of $F \approx 2$ to one of $1+(F-1)/M$. The results using this technique in Table 2, Figure 2 and in Appendix I are essentially indistinguishable from fully conservative ones, using $M = 8$, which leads to a small, approximately 20% slowdown compared to a direct-force, non-conservative trajectory. Appendix I contains further explanation of the MTS technique, as well as more detailed results for MTS simulations using models trained on the water and OC20 datasets. The technique can be also successfully used for constant-pressure simulations, as shown in Appendix G.3.

## 5. Discussion

Chemical and materials modeling is at the forefront of development in the applications of machine learning to science. The field has long been advocating for the use of physically informed architectural constraints, but there are indications that its bitter-lesson moment is coming, with the realization that deploying physics-agnostic models at scale provides better outcomes than exploiting physical priors. It appears that this is the case for some of the geometric symmetry constraints (Pozdnyakov & Ceriotti, 2023; Neumann et al., 2024; Qu & Krishnapriyan, 2024), and a growing number of architectures disregard the physical connection between the interatomic potential and the corresponding forces (Gasteiger et al., 2021; Neumann et al., 2024; Liao et al., 2023). With respect to this latter constraint, our study paints a nuanced picture. Atomistic simulations rely on the assumption that forces are the exact derivatives of the potential, and small deviations from this constraints lead to instabilities. Non-conservative behavior also results in molecular dynamics trajectories exhibiting a spontaneous drift away from the desired thermodynamic conditions. Controlling this effect with thermostats requires careful tuning, and disrupts both time-dependent properties and the sampling efficiency of the trajectory, negating the computational advantage of a direct-force architecture.

Contrary to the case of rotational symmetry that is easy to monitor and correct at inference time (Langer et al., 2024), and learn through data augmentation, assessing non-conservative behavior requires the explicit evaluation of the Jacobian both as diagnostics and as additional loss term. Furthermore, energy and forces are complementary targets, and disregarding the former may lead to potentials that appear stable, resilient to dataset inconsistencies, and with good

validation set accuracy, but are less reliable in describing the slow, collective structural rearrangements that are often the key drivers of the most relevant microscopic processes.

Given that the target forces are conservative, accurate models usually exhibit less pronounced non-conservative behavior. As a consequence, one can expect that, as the field moves to larger training datasets and more expressive models, some of the pathological effects we observe will become less severe. Our findings, however, suggest that the best way to exploit the speed-up afforded by direct prediction of the forces is not to replace conservative models, but to augment them with a non-conservative head. This can be used to accelerate training by first training a non-conservative model and then fine-tuning its energy head to yield accurate conservative forces through differentiation. The resulting "multi-force" models can also be used to speed up many different types of simulations, by alternating conservative and non-conservative evaluations, avoiding the narrower applicability and inherent instability associated with relying exclusively on direct force predictions. This insight enables the efficient training of the next generation of universal machine-learning interatomic potentials while retaining the physical correctness required for practical simulations.

## Software and Data

Code and data required to reproduce the results in this work are available on Zenodo at https://zenodo.org/records/14778891. An example of how to perform multiple-time-step dynamics with conservative and direct forces can be found at https://atomistic-cookbook.org/examples/pet-mad-nc/pet-mad-nc.html, and an example of conservative fine-tuning at https://atomistic-cookbook.org/examples/pet-finetuning/pet-ft-nc.html. More details on the software used are available in Appendix J.

## Acknowledgements

The authors would like to thank Federico Grasselli and Niklas Schmitz for stimulating discussion, and Rafael Gomez-Bombarelli for a coffee-break conversation which inspired us to look into this problem. ML and MC acknowledge funding from the European Research Council (ERC) under the European Union's Horizon 2020 research and innovation programme Grant No. 101001890-FIAMMA. FB and MC acknowledge support from the NCCR MARVEL, funded by the Swiss National Science Foundation (SNSF, grant number 182892) and from the Swiss Platform for Advanced Scientific Computing (PASC).

## Impact Statement

This paper presents work whose goal is to advance the application of machine learning to simulations of molecules and materials, which can be used for many purposes. As potential impacts are hard to predict and wide-ranging, we do not believe it is necessary to discuss them here in detail.

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

## A. Forces and potentials as training targets

Including both forces and the potential energy $V$ as targets when training a ML potential (either separately, or jointly for a conservative model) requires choosing a weighting factor to combine the errors into a single loss function. Independently from the relative weight, it is interesting to consider how these two targets affect the accuracy of a model for different types of molecular displacements. To investigate this aspect, we can approximate a material in the vicinity of a stable structure, i.e., a local minimum of the potential energy $V$, as a quadratic form, which can be written in the basis of the eigenvectors of the (mass-scaled) Hessian to give a superimposition of harmonic modes,

$$V(\mathbf{q}) = \sum_k V_k(q_k) = \frac{1}{2} \sum_k m\omega_k^2 q_k^2, \tag{8}$$

where $m$ is the atomic mass (the expression generalizes to the case of multiple atomic types), $\omega_k$ are normal mode frequencies and $q_k$ the displacements along the eigenvectors.

If we now consider how configurations are sampled at a constant temperature $T$ (running short MD trajectories is a common strategy to generate training sets for MLIPs), one sees that each harmonic mode is distributed as $p(q_k) \propto \exp(-V_k(q_k)/k_\mathrm{B}T) = \exp(-m\omega_k^2 q_k^2/2k_\mathrm{B}T)$. As a consequence, one can easily compute the expectation values

$$\langle q_k^2 \rangle \propto \frac{k_\mathrm{B}T}{\omega_k^2}, \quad \langle V_k(q_k) \rangle \propto k_\mathrm{B}T, \quad \langle f_k^2 \rangle \propto k_\mathrm{B}T\omega_k^2 \tag{9}$$

These textbook results highlight the following facts: (1) low-frequency normal modes are those associated with the largest structural deformations – and therefore, often, with phase transitions and important molecular rearrangements; (2) thermal excitations affect all normal modes equally in terms of potential energy contributions; (3) the largest force contributions come from high-frequency (low-displacement) vibrations.

Thus, when training on forces using an $L^2$ loss, molecular modes associated with high-frequency vibrations are over-emphasized. For example, if one considers the water dataset we use in this work (Cheng et al., 2019), the total force acting on each water molecule has a root mean square of 0.97 eV/Å, while the residual "intra-molecular" forces (that are predominantly short-ranged and associated with high-frequency molecular vibrations) are four times larger, 3.93 eV/Å. This very crude analysis highlights the non-trivial implications of using forces as (direct or indirect) training targets.

## B. Theoretical computational cost

Conservative forces are generally computed from potential energies by backward propagation of gradients. The vast majority of operations in neural networks (and nearly all those that take up significant computational time) are binary operations which can be expressed as the computation of $f(x, y)$ starting from $x$ and $y$. During the backward step corresponding to such operation, $\partial V/\partial x$ and $\partial V/\partial y$ must be found from $\partial V/\partial f$. In the case of matrix multiplication, the forward calculation of $f(x, y)$, the backward calculation of $\partial V/\partial x$ and the backward calculation of $\partial V/\partial y$ each consist of a matrix multiplication with the same computational cost. Since matrix multiplications can be assumed to be the most costly components of neural networks, this generally implies that backward gradient computations are around twice as expensive as the corresponding forward function evaluation.

However, in the case of backward force evaluation, operations where $x$ is an internal representation and $y$ is a weight can save the $\partial V/\partial y$ calculation. In a simple multi-layer perceptron, where all linear layers correspond to this type of computation, this would yield a backward propagation that is roughly as expensive as the forward pass. This is not the case in transformers, as the attention mechanism involves comparatively expensive operations where both $x$ and $y$ are internal representations, and one can expect the backward propagation of gradients to be somewhere between $1\times$ and $2\times$ the cost of the forward evaluation.

## C. Range of back-propagated and direct force models

The use of a cutoff to restrict the range of interactions is ubiquitous in the construction of physics-based potentials, and is also an integral part of descriptor-based ML potentials (Musil et al., 2021). It is often argued that models that incorporate correlations between at least two neighbors of each central atom achieve an effective interaction range of twice the cutoff distance (Artrith et al., 2011) (see Figure 5a). A similar effect also applies to message-passing architectures (Nigam et al., 2022).

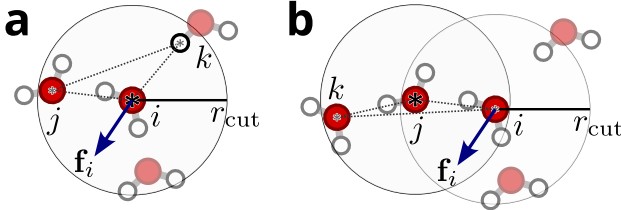

*Figure 5.* A schematic representation of the interactions in a (ML) interatomic potential. (a) Contributions to an $i$-centered prediction of energy or force from three or more neighbors, labeled $j$ and $k$. (b) Contributions from a $j$-centered energy prediction to the force on the $i$-th atom, computed by back-propagation.

One can see clearly that this extension of the interaction range beyond the cutoff (or the receptive radius of the GNN) does not apply to the case in which forces are predicted directly. Considering for simplicity the case of a three-body model in which the potential contribution from each atomic environment $i$ can be written as a sum of a function of its distances with two neighbors $j$ and $k$, and the distance between the neighbors $r_{jk}$

$$V_i = \sum_{j,k} v(r_{ij}, r_{ik}, r_{jk}),\tag{10}$$

one sees that the dependency on interatomic distances greater than the cutoff is due to the relative position of atoms other than the central atom. Thus, a direct model that predicts a similar 3-body force

$$\mathbf{f}_i = \sum_{j,k} \mathbf{f}(r_{ij}, r_{ik}, r_{jk}),\tag{11}$$

or any other functional form limited to the neighbors of the $i$-th atom, only contains information on atoms within the cutoff. The dependency of $\mathbf{f}_i$ on the coordinates of far-away atoms is a consequence of the fact that the total energy is built as a sum over multiple centers. It is only through the terms of the form $\partial V_j / \partial \mathbf{r}_i$ that occur naturally when evaluating forces through back-propagation, than the force depends on the position of the neighbor's neighbors. More generally, in a message-passing implementation, backpropagation ensures that force predictions benefit from an effective receptive radius that is twice that of atom-centered energy predictions.

A further concern for direct force models is that – at least for low-body-order models – atom-centered descriptors can be shown to have low resolving power, with pairs of distinct atomic environments having precisely the same representation (Pozdnyakov et al., 2020). For all known degeneracies, descriptors centered on other atoms allows distinguishing the structure *as a whole*, and as a consequence the total energy and interatomic forces can be still differentiated. This would not be the case for direct force predictions, which would fail completely to differentiate degenerate pairs when using atom-centered low-body-order models, and would be more sensitive to numerical instabilities for higher-order models.

In practice, we find consistent evidence of the practical impact of these considerations. As an example, Table 3 reports the accuracy of PET-C and PET-NC models, using 2 and 3 message passing layers. It can be seen that the accuracy of the direct force model benefits much more from the increase in the receptive radius.

| Model | 2 message-passing layers | 3 message-passing layers |
|---|---|---|
| PET-C | 20.8 | 18.6 |
| PET-NC | 32.8 | 24.3 |

*Table 3.* Test-set force MAE, in units of meV/Å, for conservative and non-conservative force-only PET models using 2 and 3 message-passing layers, respectively. Upon increasing the receptive radius, the accuracy of the NC model improves by around 30%, while the accuracy of the C model only improves by around 10%.

## D. Models and architectures

Even though the main text is focused on custom-trained models, and emphasize the most accurate non-conservative model we could obtain, we also want to provide an overview of the behavior of a less-performant custom-trained model, and of several publicly available general-purpose models. A comprehensive list of the models we consider is reported in Table 4.

| Model | Description |
|---|---|
| ORB-v2 | Non-equivariant, non-conservative model, trained on the Alexandria and MPtrj datasets. |
| Equiformer | Equivariant, non-conservative model, trained on the Alexandria and MPtrj datasets. |
| MACE-MP-0 | Equivariant, conservative model, trained on the MPtrj dataset. |
| SevenNet | Equivariant, conservative model, trained on the MPtrj dataset. |
| PET-C | A re-implementation of the PET architecture, trained on the bulk water dataset. |
| PET-NC | A modified PET architecture, trained on the bulk water dataset. |
| SOAP-BPNN-C | A SOAP-BPNN architecture, trained on the bulk water dataset. |
| SOAP-BPNN-NC | A modified SOAP-BPNN architecture, trained on the bulk water dataset to directly predict forces. |

*Table 4.* Models used in the present work.

It should be noted that:

- The ORB model (orb-v2) is more accurate than MACE-MP-0 and SevenNet, as the former is pre-trained on the Alexandria (Schmidt et al., 2024) dataset and then fine-tuned on MPtrj, while the latter two are only trained on MPtrj. Despite its higher accuracy, ORB yields problematic physical behavior as discussed in this work.

- The PET-NC and SOAP-BPNN-NC models are simply obtained from the respective conservative models by changing the output head to predict atomic forces directly. In the case of SOAP-BPNN-NC, an equivariant vector representation is generated internally thanks to the formalism in Villar et al. (2021). In both cases, due to the marginal increase in number of parameters in the force head, the non-conservative models have slightly more parameters than their conservative counterparts. Within the calculators for these two non-conservative models, we implemented the net force removal suggested in Neumann et al. (2024).

The architectures of these models are further described here:

- ORB: A rotationally unconstrained and non-conservative architecture, presented in Neumann et al. (2024).

- MACE: A rotationally invariant and conservative architecture, presented in Batatia et al. (2022).

- SevenNet: The SevenNet model (Park et al., 2024) makes use of the NequIP (Batzner et al., 2022) architecture, which is rotationally invariant and conservative.

- PET-C and PET-NC: A re-implementation of the architecture in Pozdnyakov & Ceriotti (2023), which is rotationally unconstrained and conservative. The non-conservative version changes the final head to predict forces instead of energies.

- SOAP-BPNN-C and SOAP-BPNN-NC: A Behler-Parrinello neural network architecture (Behler & Parrinello, 2007), using SOAP (Bartók et al., 2013) descriptors. This architecture is rotationally invariant and conservative. The non-conservative version makes use of the formalism in Villar et al. (2021) to predict forces (a vector) from a scalar internal representation.

### D.1. Timings of general-purpose models

Table 5 shows the timings for the four general-purpose models tested in this work, compared with the PET models we train here. The large version of MACE-MP-0 was used in this table and throughout this work. The small version of EquiformerV2 was used in this table and throughout this work, except to calculate work loops, where the large version was used.

*Table 5.* Timings (ms) of general-purpose models on a bulk water NVE simulation with 64 water molecules (192 atoms), on an Nvidia H100 GPU. Note that timings refer to the overall evaluation time for a single configuration, performed using the ASE calculator distributed with each model, and therefore may contain overheads that are not directly associated with the model evaluation (neighbor list construction, etc.). The timing of the water-focused PET-C and PET-NC are also reported, for reference.

| MODEL | TIMING PER STEP | TIMING PER STEP PER ATOM |
|---|---|---|
| MACE (C) | 26.9 | 0.140 |
| SEVENNET (C) | 52.8 | 0.275 |
| ORB (NC) | 11.9 | 0.062 |
| EQUIFORMER (NC) | 1580 | 8.230 |
| PET (C) | 19.4 | 0.101 |
| PET (NC) | 8.58 | 0.047 |

| ARCHITECTURE | TYPE | TRAINING | MAE($V$) | MAE($\mathbf{f}$) | TIMING (TR.) | TIMING (EV.) |
|---|---|---|---|---|---|---|
| PET | – | $V$ | 4.7 | 1025.6* | 5.48 | 0.0264 |
| PET | C | $\mathbf{f}$ | 1.26** | 18.6 | 15.30 | 0.0713 |
| PET | C | $V, \mathbf{f}$ | 0.55 | 19.4 | 15.31 | 0.0716 |
| PET | NC | $\mathbf{f}$ | – | 24.3 | 5.55 | 0.0224 |
| PET | NC | $V, \mathbf{f}$ | 1.42 | 24.8 | 5.63 | 0.0269 |
| PET-M | C | $V, \mathbf{f}$ | 0.59 | 20.2 | 15.43 | 0.0715 |
| PET-M | NC | | | 26.7 | | 0.0265 |
| PET-M-FT*** | C | $V, \mathbf{f}$ | 0.50 | 20.0 | 56.42*** | 0.0714 |
| SOAP-BPNN | – | $V$ | 2.16 | 177.0* | 3.57 | 0.1065 |
| SOAP-BPNN | C | $\mathbf{f}$ | 1.89** | 40.6 | 36.10 | 0.6394 |
| SOAP-BPNN | C | $V, \mathbf{f}$ | 1.38 | 41.4 | 36.20 | 0.6367 |
| SOAP-BPNN | NC | $\mathbf{f}$ | – | 112.2 | 5.81 | 0.1515 |
| SOAP-BPNN | NC | $V, \mathbf{f}$ | 3.20 | 111.9 | 6.34 | 0.1674 |

*Table 6.* Test errors (energies in meV per atom, forces in meV/Å), training and evaluation timings for models trained on the bulk water dataset. Training timings correspond to the time to compute a single epoch (in seconds) on 4 H100 GPUs with a total batch size of 64. Evaluation timings correspond to the average time per atom (in ms) for energy and/or force evaluations for single structures across the test set.

*The force errors of energy-only models are computed by evaluating forces as derivatives of the energies, despite the fact that no explicit force training took place.

**A linear fit was executed to minimize training errors on energies for the force-only models, in order to calculate the best constant shift for the fictitious energy of which the forces are the derivatives.

***Trained on a single GPU as opposed to four, and for a single day as opposed to two.

### D.2. Accuracy and timings of water models

Similarly to Table 1, we evaluate the accuracy of the SOAP-BPNN architecture on the same bulk water dataset under different training conditions. The results in Table 6 show, once again, that the lack of energy conservation without a corresponding data augmentation strategy seems to hurt the accuracy of the models.

In general, although the non-conservative models trained in this work on the bulk water dataset (with little more than 250000 targets) seem to show worse accuracy, training on larger datasets has shown that non-conservative models can be competitive in accuracy with conservative models. This is not only because energy conservation can then be effectively learned, but also because non-conservative models, by virtue of being faster, can train for a larger number of epochs at a given computational budget. Training duration can be the limiting factor to accuracy on large datasets.

The timings of the PET models are fully consistent with the theoretical cost analysis in Appendix B. In contrast, the SOAP-BPNN models rely on the SOAP atomic descriptors as implemented in `https://github.com/metatensor/featomic`. Within this library for atomistic descriptors, three implementation details account for the SOAP-BPNN timings: 1) although the models are trained and evaluated on GPU, the feature calculation is executed on CPU; 2) feature calculation is parallelized across different structures, but not different chemical environments within the same structure (effectively meaning that no evaluation-time parallelization is present); 3) the equivariant calculation of forces makes it necessary to evaluate additional features with an angular momentum quantum number of 1.

## E. Non-conservative work

Since non-conservative models do not obey equation (5), we evaluate indicative magnitudes of the work over a closed loop for the non-conservative models considered in this study. These are shown in Figure 6.

The closed path corresponds to the rotation and deformation of a single water molecule within a liquid water structure, while keeping all the other atoms fixed. Figure 6 shows the cumulative work of the models considered in this study. Even though the cumulative work curves are very similar, the conservative models results in zero overall work on the closed path; meanwhile, the non-conservative models leads to an overall non-zero work along the path.

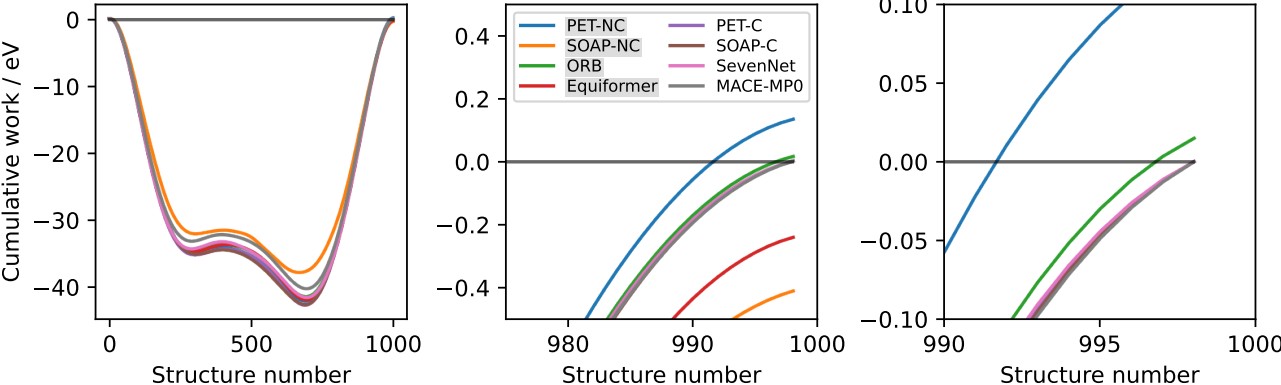

*Figure 6.* Cumulative work along a closed path for all models considered in the study. While the total work for the conservative models is zero up to machine precision, the non-conservative models exhibit a non-zero total work (ORB: 15 meV, Equiformer: -241 meV, PET-NC: 132 meV, SOAP-BPNN-NC: -410 meV). The first figure from the left shows the overall path, while the other two zoom in on the last part.

## F. Geometry optimization

### F.1. Optimization trajectories

We compare the behavior of different models when quenching a liquid-water snapshot, optimizing it towards the nearest potential energy minimum, (Figure 7) showing both the convergence in terms of the force modulus, and the trajectory of configurations as a latent-space projection built on geometric descriptors. The latent-space plots are obtained by computing SOAP descriptors for all configurations in all the trajectories, averaged over atomic centers in each structures, and are projected on the axes of highest variability using simple Principal Component Analysis (PCA). The qualitative features of the latent space plots are insensitive to the details of the SOAP descriptors used.

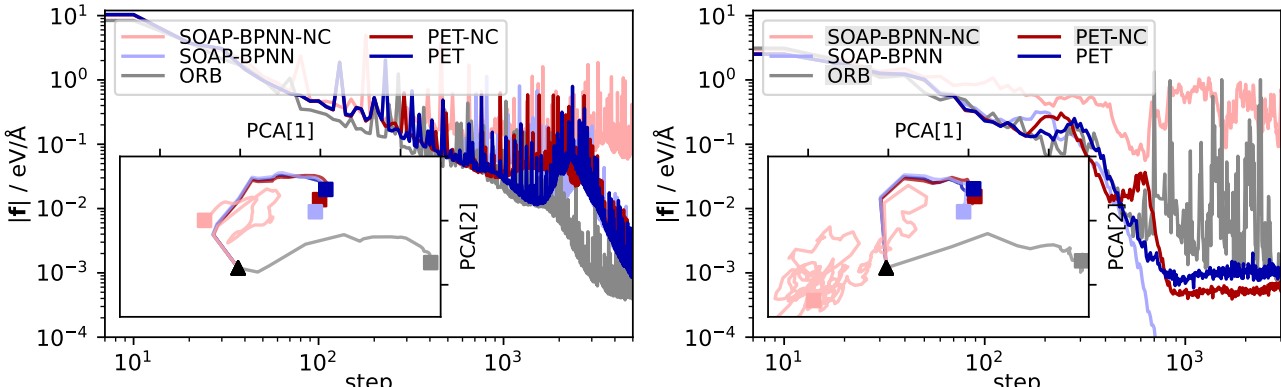

*Figure 7.* Force modulus as a function of step along FIRE (left) and BFGS (right) optimization trajectories of a liquid water snapshot, for different MLIPs. The inset shows a latent-space projection of the trajectory in configuration space; the triangle indicates the starting configuration, and the squares the final one

We discuss first the results for the FIRE algorithm (left panel in Figure 7). Despite the very different level of accuracy, the conservative SOAP-BPNN model and PET converge to a similar configuration (with PET forces saturating at about $10^{-3}$ eV/Å, due to numerical precision). The non-conservative PET-NC model also converges to a similar structure, indicating that a sufficiently accurate non-conservative model can indeed be used with a gradient-based structural optimization algorithm. The non-conservative SOAP-BPNN model, however, displays a catastrophic mode of failure, with the force never decreasing below 0.1 eV/Å, and the trajectory drifting off in a different direction without ever reaching a stable state. ORB converges to a very different structure than the other models, which might be due to the different reference DFT functional used for training.

The BFGS optimization trajectories (right panel in Figure 7) are very similar to those obtained with FIRE, except for SOAP-BPNN-NC whose catastrophic failure is apparen in the trajectory going in a completely different direction, and in the force modulus never converging below 0.1 eV/Å. Thanks to the second-order nature of LBFGS, convergence is much faster, until PET-based models reach their precision limit. ORB shows an interesting behavior, as it reaches a similar configuration as with FIRE, but the force modulus exhibits large fluctuations, and it would be hard to determine a clear threshold to establish convergence. On a practical level, this experiment shows that non-conservative forces make geometry optimization more fragile – with first-order methods being somewhat more stable although slower – and require careful choice of the minimization algorithm and its convergence parameters.

### F.2. Failure rates

In Table 7, geometry optimization is attempted with a range of conservative and non-conservative models. For each model, three cases are considered: 1) geometry optimization of gas-phase water molecules, starting from the experimental geometry, randomly displacing the coordinates with a standard deviation of 0.5 Å, and relaxing the geometry; 2) geometry optimization of bulk water structures from the test set of (Cheng et al., 2019), 3) geometry optimization of molecules chosen at random from the QM9 dataset (Ramakrishnan et al., 2014), randomly displacing the coordinates with a standard deviation of 0.5 Å before relaxing the structures. Geometry optimization is performed with the L-BFGS (Liu & Nocedal, 1989) algorithm as implemented in ASE (Hjorth Larsen et al., 2017). Optimization runs that do not converge within 1000 optimization steps are considered as failed.

*Table 7.* Percentage success rate in geometry optimization.

| MODEL | $H_2O_{(g)}$ | $H_2O_{(l)}$ | QM9 | MPTRJ |
|---|---|---|---|---|
| ORB-LOW-PRECISION* (NC) | 3 | 0 | 0 | 10 |
| ORB (NC) | 69 | 9 | 1 | 76 |
| SEVENNET (C) | 81 | 88 | 92 | 97 |
| MACE (C) | 94 | 83 | 94 | 99 |
| PET-NC (NC) | 75 | 52 | – | – |
| PET-C (C) | 83 | 58 | – | – |
| SOAP-BPNN-NC (NC) | 79 | 0 | – | – |
| SOAP-BPNN-C (C) | 91 | 59 | – | – |

*This is an ORB model used with its default settings, which lower the precision of matrix multiplications. Given the results here, we deactivated this setting for all ORB results shown in the rest of this work.

Non-conservative models consistently show lower rates of success in geometry optimization. It should be noted that strict convergence criteria were used (`fmax=1e-5`Å for molecules and `fmax=1e-4`Å for bulk systems).

# G. Additional molecular dynamics results

### G.1. NVE temperature profiles

We repeat the experiments in Sec. 4.4 using all the potentials we tested. As shown in Figure 8, all conservative models yield a stable kinetic temperature profile (there is a spread in the average temperature that is compatible with the nature of constant-energy trajectories) and all the non-conservative models show noticeable temperature drift. Equiformer displays a drift comparable to PET-NC, which is remarkable for a general-purpose model (but is too expensive to be used in practical simulations). SOAP-BPNN-NC shows an extremely large drift, consistent with the fact it is a very inaccurate model.

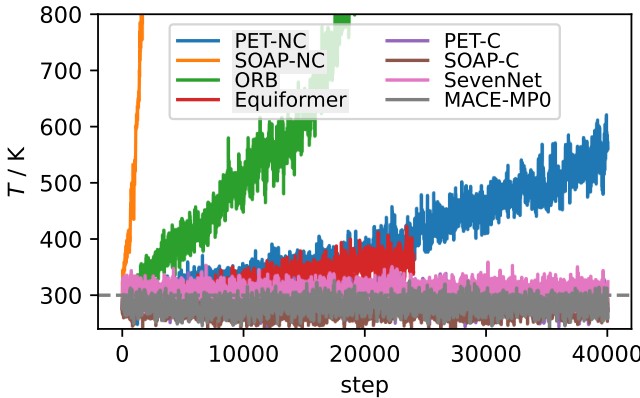

*Figure 8.* A thorough comparison of kinetic temperature profiles for NVE MD trajectories performed with different conservative (right column in the legend) and non-conservative (left column in the legend) ML potentials.

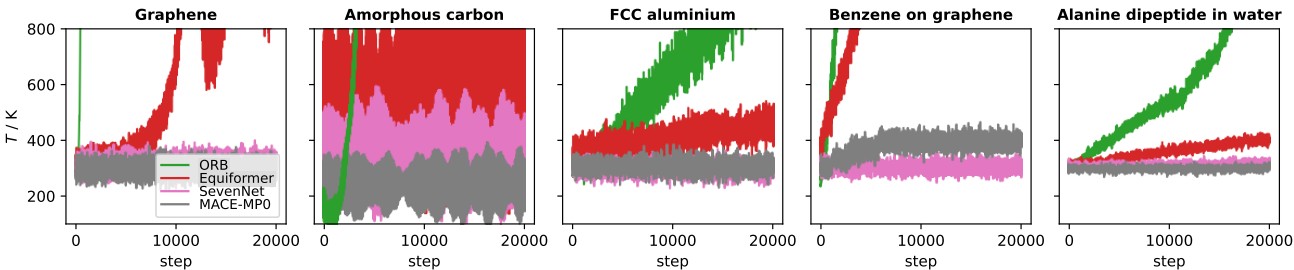

*Figure 9.* Kinetic temperature profiles for NVE MD trajectories, using three universal models, for three different materials, as indicated. Simulations for graphene and amorphous carbon were performed with a time step of 1 fs, and those for aluminum with a time step of 2 fs, reflecting the different timescale of atomic motion.

We also perform NVE MD trajectories on a wider range of chemical systems using the general-purpose potentials MACE-MP, SevenNet, Equiformer and ORB. Fig. 9 shows that the non-conservative models ORB and Equiformer exhibit a strong kinetic energy gain during the simulation of graphene, amorphous carbon and FCC aluminium, with ORB exhibiting an explosive behavior for graphene (possibly because it is outside the training set). The conservative models exhibit stable temperature fluctuations, even for amorphous carbon where the high distortion of the structure leads to higher initial energy and broader kinetic energy fluctuations. The order of magnitude of the drift is similar for systems with different degrees of chemical heterogeneity.

## G.2. NVT

We repeated the bulk water experiments in Sec. 4.5 with a wider range of models and thermostats. The results are summarized in Table 8.

*Table 8.* Average kinetic temperatures (in K) for different models and thermostatting schemes. For SVR, we show also the separate kinetic temperature computed only for hydrogen and oxygen atoms. All thermostat time constants are reported in units of fs.

| THERMOSTAT TYPE THERMOSTAT $\tau$ | WN 10 | WN 100 | WN 1000 | SVR 10 | SVR 10 | SVR 10 |
|---|---|---|---|---|---|---|
| ATOMS | ALL | ALL | ALL | ALL | H | O |
| ORB (NC) | $300.4_{(0.1)}$ | $304.2_{(0.2)}$ | $351.0_{(0.6)}$ | $301.0_{(0.1)}$ | $336.2_{(0.8)}$ | $229.5_{(1.6)}$ |
| MACE (C) | $300.1_{(0.1)}$ | $300.4_{(0.2)}$ | $300.1_{(0.4)}$ | $300.1_{(0.1)}$ | $299.0_{(0.6)}$ | $302.3_{(1.1)}$ |
| SEVENNET (C) | $300.0_{(0.1)}$ | $300.1_{(0.2)}$ | $299.9_{(0.5)}$ | $300.1_{(0.1)}$ | $299.9_{(0.4)}$ | $300.5_{(1.1)}$ |
| PET-NC (NC) | $300.1_{(0.1)}$ | $301.4_{(0.2)}$ | $312.8_{(0.5)}$ | $300.3_{(0.1)}$ | $295.6_{(0.3)}$ | $309.9_{(0.6)}$ |
| PET-C (C) | $300.1_{(0.1)}$ | $300.1_{(0.2)}$ | $300.3_{(0.7)}$ | $300.1_{(0.1)}$ | $299.6_{(0.3)}$ | $301.0_{(0.7)}$ |
| SOAP-BPNN-NC (NC) | $301.9_{(0.1)}$ | $340.7_{(0.2)}$ | $1.2 \cdot 10^6$ | $309.7_{(0.1)}$ | $265.5_{(0.2)}$ | $399.5_{(0.4)}$ |
| SOAP-BPNN-C (C) | $299.9_{(0.1)}$ | $299.8_{(0.4)}$ | $300.5_{(0.5)}$ | $300.1_{(0.2)}$ | $298.9_{(0.7)}$ | $302.4_{(1.5)}$ |

## G.3. Direct stresses: NPH and NPT

In order to test the effects of direct stress prediction, which can be employed in constant-pressure simulations, we run MD in the NPH and NPT ensembles. In this case, as a means to disentangle the effects of non-conservative stresses from those of non-conservative forces, we run the "non-conservative" (NC) models using non-conservative stresses and *conservative* forces. For these simulations, we employ the PET-MAD general-purpose potential (Mazitov et al., 2025), whose version 1.1 is trained to provide both conservative and non-conservative stresses.

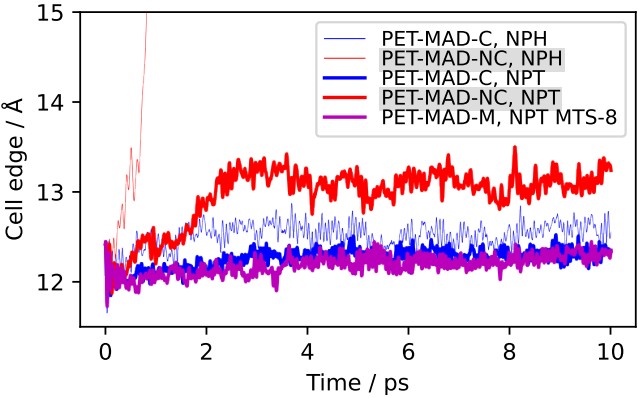

*Figure 10.* Length of the cubic cell edge, as a function of time, during NPH and NPT simulations of liquid water using the PET-MAD model, in conservative mode, non-conservative mode, and using multiple time stepping with $M = 8$. The Bussi-Zykova-Parrinello barostat (Bussi et al., 2009) was used in all simulations.

Fig. 10 shows how the conservative runs produce reasonable cell volumes, both in the constant-enthalpy NPH ensemble and the isothermal-isobaric NPT ensemble. The non-conservative NPH run shows rapid and unphysical expansion of the cell, which is reflected in the rapid growth in the enthalpy of the simulation (which is instead conserved very well by the conservative NPH run). This "explosion" of the cell is prevented by the thermostat in the NPT ensemble, although the exact value of the cell parameter is inconsistent with the conservative NPT baseline. Finally, we show a MTS (multiple time stepping, see App. I) run using $M = 8$, which correctly reproduces the cell volume of the conservative NPT simulation while only evaluating the more expensive conservative stresses every 8 time steps. This simple example illustrates how all the considerations that were made for non-conservative forces in the context of NVE/NVT simulations are also applicable to NPH/NPT simulations when using direct stress predictions.

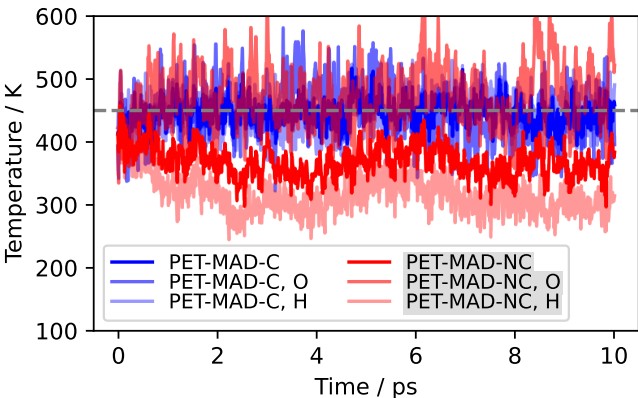

*Figure 11.* Total and atom-type-resolved kinetic energy profiles for NVE simulations of liquid water using the conservative and non-conservative modes of the PET-MAD (Mazitov et al., 2025) potential. In the non-conservative case, energy conservation is enforced by velocity rescaling.

## G.4. Energy conservation by velocity rescaling

The easiest way to stabilize NVE molecular dynamics using direct forces is to rescale the velocities (or, equivalently, the momenta) at each step to enforce conservation of energy, as proposed in Bigi et al. (2025). While this leads to an improvement, in the sense that NVE trajectories are guaranteed to be stable, many fundamental issues remain. As an example, Fig. 11 compares traditional NVE MD with a conservative model to NVE MD with a non-conservative model, where the latter is stabilized by velocity rescaling. The conservative model shows kinetic temperatures close to the temperature at which the starting structure was equilibrated, both for the overall simulation and for individual atomic types. In contrast, the non-conservative run departs from the equilibration temperature and, more importantly, it shows much different average kinetic energies for atoms belonging to different chemical species. This behavior is in violation of the principle of equipartition of energy, and it is a manifestation of the fact that the target ensemble (NVE, in this case) is not being sampled correctly. This is very similar to what we observe with global thermostats, and is an analogous manifestation of the fact that enforcing the correct values of global quantities does not guarantee that all the parts of a simulation will equilibrate correctly, in the absence of a well-defined Hamiltonian.

### G.5. Structural correlations

Even though the kinetic temperature can be used as a diagnostic of the ensemble error, it does not guarantee that the trajectory yields structural properties consistent with the correct NVT ensemble. We compute different kinds of structural correlation functions: the O-O and H-H pair correlation functions $g_{OO}(r)$ and $g_{HH}(r)$ reports on the probability of finding two oxygen or two hydrogen atoms at a given distance $r$, and the orientational correlation function $c_{uu}(r)$ reports the average scalar product between the orientation vector of two water molecules at a given distance. Results for the PET models (Figure 12) show that structural correlations are consistent for all thermostatting schemes for the conservative runs. The 10 fs white-noise Langevin runs exhibit larger error bars, consistent with the poor sampling efficiency. All non-conservative PET-NC runs show statistically significant discrepancies with respect to the conservative reference. Although it is not possible to rigorously disentangle sampling errors due to non-conservative forces, and the overall difference in the force values between PET and PET-NC, we hypothesize that the differences seen in the strong-Langevin-coupling limit are due to differences in the underlying forcefield, while the differences for the weaker-coupling $\tau = 1$ ps and global SVR thermostat runs are associated with the deviation in the sampling temperature.

For the ORB models (Figure 13) we consider the high-coupling limit as the most reliable reference of the expected correlations with correct sampling. The large difference in reference correlations with respect to the custom-trained PET models is due to the difference in reference DFT energetics: the PBEsol functional used in MPTraj is known to yield severely overstructured water, while the revPBE0-D3 functional used in (Cheng et al., 2019) was shown to give good agreement with experimental data. For the purpose of this work, however, this difference is not important, and the main takeaway from Figure 13 is that for ORB there are large discrepancies between different thermostatting schemes, which is indicative of the artifacts induced by non-conservative forces, and the difficulty in mitigating them using thermostats.

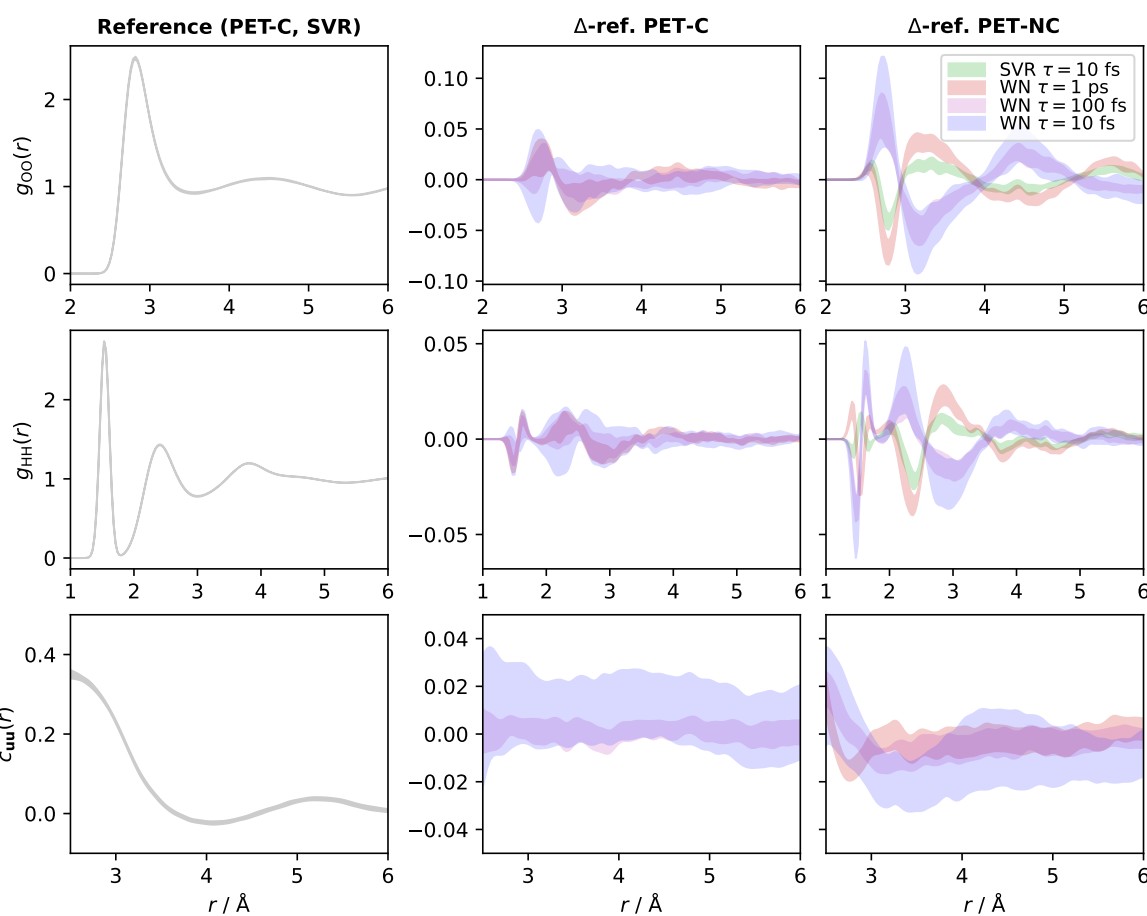

*Figure 12.* Structural correlations for simulations of room-temperature water using different PET models. From top to bottom, the rows report the O-O correlation function, the H-H correlation function, and the orientational correlation function. The left column shows the reference value (SVR thermostat for the conservative PET model), the middle column the differences with respect to this reference for PET-C using different thermostats, and the right column differences to the reference using the non-conservative PET model and different thermostats. Shaded areas encompass two standard errors around the mean values.

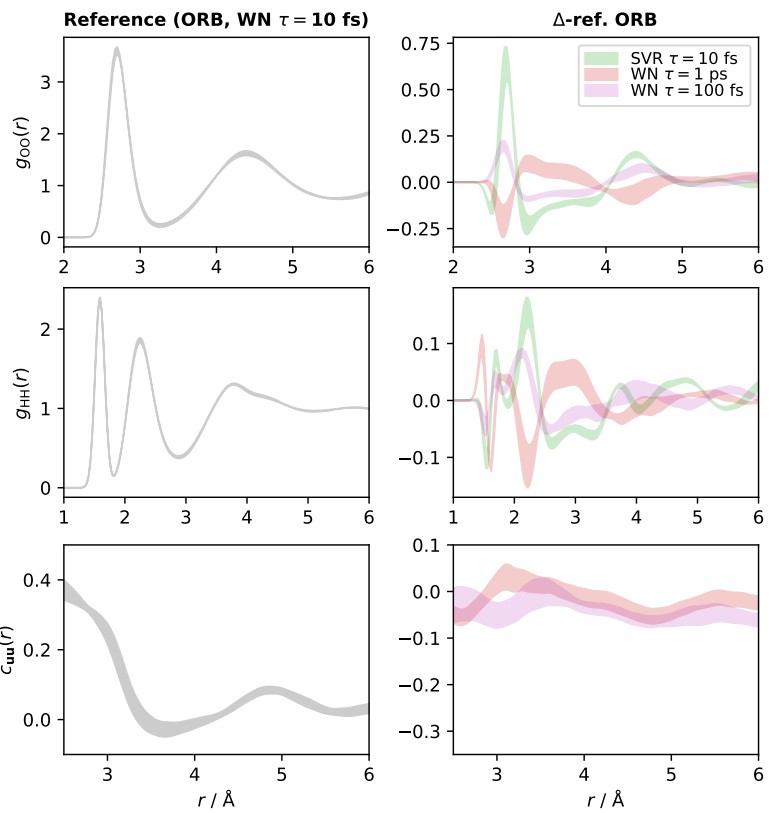

*Figure 13.* Structural correlations for simulations of room-temperature water using the ORB model and different thermostatting scheme. From top to bottom, the rows report the O-O correlation function, the H-H correlation function, and the orientational correlation function. The left column shows the reference value (strong-coupling, $\tau = 10$ fs white-noise Langevin thermostatting) and the right column the differences with respect to this reference for ORB trajectories using different thermostats. Shaded areas encompass two standard errors around the mean values. Note the much larger range of the discrepancies with respect to the PET-NC runs in Figure 12.

## G.6. Generalized Langevin equations

The generalized Langevin equation (GLE) thermostat (Ceriotti et al., 2010) is a local stochastic thermostat that extends traditional Langevin dynamics by introducing a history-dependent friction term. The possibility of tuning the memory kernel of the friction and the noise allows tuning the effect of the thermostat, by effectively coupling different modes of the physical system to different friction timescales. Thanks to this construction, near-optimal damping can be achieved for a large range of characteristic frequencies, as opposed to the traditional Langevin thermostat, which only achieves optimal damping (and therefore sampling) for a narrow frequency range.

Among the many different applications of GLE thermostats, that of Ref. 50 appears particularly suited to contrast the temperature drift observed with the direct prediction of forces: by applying a high-pass filter to the noise, it allows to thermostat aggressively the fast degrees of freedom of the system, without disrupting the long-time dynamics and the sampling of the diffusive degrees of freedom. We use the same parameters that have been used successfully to control temperature drift caused by the integration errors for aqueous systems in Ref. 50, using $1/\gamma_0 = 83.33\,\mathrm{ps}$, $1/\gamma_\infty = 10\,\mathrm{fs}$, and $\omega_F = 300\,\mathrm{ps}^{-1}$, where all GLE parameters refer to their definition in Morrone et al. (2011).

We probe the behavior of conservative and non-conservative models under GLE-thermostatted dynamics. Table 9 shows that this GLE, that could successfully control integration errors for an empirical water model, fails catastrophically when used with non-conservative models. This can be explained by the observation that the near-diffusive modes, which are those that show the most non-conservative behavior according to the analysis in Figure 1, are almost not damped at all by this thermostat. Even though one could in principle design a low-pass filter, or use more aggressive parameters for this high-pass thermostat, the analysis performed in Ref. 50 indicates that such GLE would slow down long-time sampling, exactly as it is the case for a traditional Langevin thermostat. By reducing the sampling efficiency, this effect negates the advantages of direct force prediction.

*Table 9.* Average kinetic temperatures (in K) for different models and atomic species, for the molecular dynamics of water using a GLE thermostat.

| THERMOSTAT TYPE | GLE | GLE | GLE |
|---|---|---|---|
| ATOMS | ALL | H | O |
| ORB (NC) | $1181.5_{(5.2)}$ | $1099.2_{(4.6)}$ | $1362.5_{(7.7)}$ |
| MACE (C) | $303.1_{(1.2)}$ | $301.7_{(1.1)}$ | $306.0_{(1.6)}$ |
| SEVENNET (C) | $304.1_{(0.8)}$ | $302.6_{(1.3)}$ | $307.3_{(2.4)}$ |
| PET-NC (NC) | $524.1_{(4.6)}$ | $495.5_{(4.0)}$ | $585.5_{(5.9)}$ |
| PET-C (C) | $301.8_{(1.2)}$ | $301.6_{(1.2)}$ | $302.2_{(1.6)}$ |
| SOAP-BPNN-NC (NC) | $1.6 \cdot 10^8$ | $2.1 \cdot 10^8$ | $5.9 \cdot 10^7$ |
| SOAP-BPNN-C (C) | $301.3_{(1.6)}$ | $301.0_{(1.4)}$ | $301.9_{(1.8)}$ |

## H. Conservative fine-tuning

Figure 14 and Figure 15 show that a non-conservative model can be trained to its asymptotic accuracy with a 2-4 times smaller investment of computational resources. This is consistent with previous observations (Gasteiger et al., 2021), and one of the main reasons for which direct prediction of the forces has been proposed in previous work. More importantly, the figures also show training curves corresponding to a model that has been initially trained with direct $V, \mathbf{f}$ heads (i.e., non-conservatively), and whose $V$ head was then "fine tuned" using energy and back-propagated forces (while the non-conservative force head was also trained to avoid its degradation). This "PET-M-FT" model achieves the accuracy of a conservative model trained "from scratch" in about 1/3rd of the total training time, demonstrating that this simple fine-tuning strategy makes it possible to recover most of the speed-up afforded by direct-force training, while having the convenience and accuracy of a conservative model. It might also be possible (although technically more complicated) to alternate gradient steps using only the direct head and steps using the conservative forces to achieve a similar effect.

*Figure 14.* Training curves (showing validation MAE force error as a function of the GPU time expenditure on a NVIDIA H100) for a conservative PET-C model, for the non-conservative head of a PET-NC model, and for the fine-tuning of a conservative model initialized from the potential energy head of the PET-NC model. Training was conducted on the liquid water dataset.

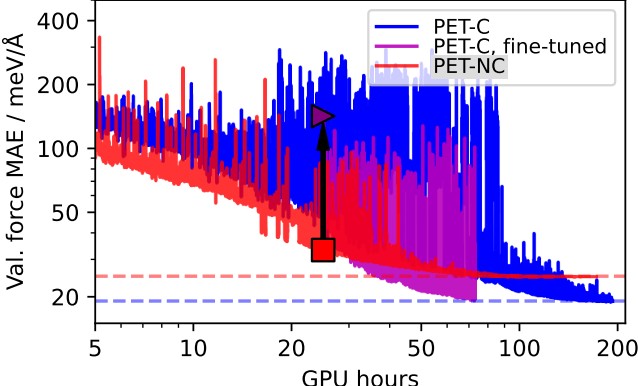

*Figure 15.* Training curves (showing validation MAE force error as a function of the GPU time expenditure on a NVIDIA H100) for a conservative PET-C model, for the non-conservative head of a PET-NC model, and for the fine-tuning of a conservative model initialized from the potential energy head of the PET-NC model. Training was conducted on a small subset of OC20 (Chanussot et al., 2021) (training on the first 15000 samples of the 200k S2EF training set, and using the next 5000 for validation).

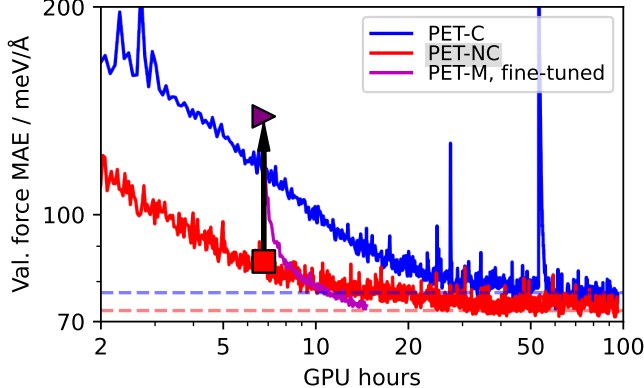

It should be noted that models trained with this fine-tuning procedure will offer accurate conservative *and* non-conservative forces, making them suitable for the multiple time stepping strategy discussed in Section 4.8. The overall procedure would allow the resulting models to be trained and evaluated at the computational cost of non-conservative models, while achieving the levels of accuracy and energy conservation of conservative models.

| Type | Relative speed (O on Sc-Co-Al alloy) | Relative speed (water) | Relative speed (theoretical) |
|---|---|---|---|
| NC | 1.0 | 1.0 | - |
| C | 1.93 | 1.92 | $F = 1.925$ |
| MTS, $M = 4$ | 1.50 | 1.45 | 1.48 |
| MTS, $M = 8$ | 1.19 | 1.18 | 1.24 |
| MTS, $M = 16$ | 1.05 | 1.04 | 1.12 |

*Table 10.* Empirically observed slowdown of conservative MD simulations for multiple time-stepping with different strides $M$, as well as a simulation where the conservative forces are evaluated at every step, compared with the theoretical cost $1 + F/M$ with $F = 1.925$. Timings are given for selected systems from the experiments in Section 4.4 and Appendix G, using runs in the NVE ensemble.

## I. Multiple time stepping

Multiple time stepping (MTS) is based on a Trotter splitting of the Liouville operator (Tuckerman et al., 1992), and can be applied any time one can decompose a potential into a "slowly varying" and expensive to compute part $V_{slow}$ and a "quickly varying", inexpensive part $V_{fast}$. The dynamics is propagated using $V_{fast}$ and a small discrete time step $\Delta t$. Every $M$ steps, one applies a correction based on $V_{slow}$. In the case of a Velocity-Verlet integrator (Verlet, 1967), a MTS integrator amounts to:

$$\mathbf{p} \leftarrow \mathbf{p} + M\mathbf{f}_{slow}\,\Delta t/2$$
$$\left.\begin{array}{l} \mathbf{p} \leftarrow \mathbf{p} + \mathbf{f}_{fast}\,\Delta t/2 \\ \mathbf{q} \leftarrow \mathbf{q} + \mathbf{p}\,\Delta t/\mathbf{m} \\ \mathbf{p} \leftarrow \mathbf{p} + \mathbf{f}_{fast}\,\Delta t/2 \end{array}\right\} \text{Repeat } M \text{ times} \qquad (12)$$
$$\mathbf{p} \leftarrow \mathbf{p} + M\mathbf{f}_{slow}\,\Delta t/2$$

where $\mathbf{q}$ and $\mathbf{p}$ indicate the vectors containing positions and momenta of all particles, and $\mathbf{m}$ the masses. Forces have to be recomputed if they are needed after a position update. In the limit of ideal time scale separation, the resulting dynamics (and the sampled ensemble for constant-temperature simulations) is consistent with $V_{slow}$, at a much-reduced cost: Letting $F$ denote the slowdown of "slow" over "fast" forces, and assuming that an evaluation of "slow" forces also yields the "fast" ones, MTS reduces the theoretical cost of well-behaved molecular dynamics from $F$ to $1 + (F - 1)/M$. Choosing $M$ therefore requires a trade-off between accuracy and speed. Morrone et al. (2011) discusses the failure modes of MTS at high $M$, and shows how to use thermostats for mitigation.

Here we use the implementation in i-PI (Kapil et al., 2016; Litman et al., 2024), and use the non-conservative forces as the "fast" forces, $\mathbf{f}_{fast} = \mathbf{f}_{NC}$, and the difference between conservative and non-conservative forces as a slow correction, $\mathbf{f}_{slow} = \mathbf{f}_C - \mathbf{f}_{NC}$. Our current implementation does not reuse computation between the conservative and non-conservative forces, and so the practical overhead is given by $1 + F/M$, with an empirical factor of $F \approx 2$ (see Appendix B). As seen in Table 10, empirical timings match the theoretical ratios rather closely.

NVE MD trajectories based on the MTS algorithms are stable up to a time step factor of $M = 8$, while trajectories with $M = 16$ become unstable and show a large kinetic temperature drift (Figure 16). Note that the cause of drift here is different than for non-conservative models, and is associated with resonances between the natural dynamics of the fast degrees of freedom and the integration errors due to imperfect time scale separation. Running at $M = 8$ recovers almost in full the speed-up associated with the use of a non-conservative force head, while allowing to run stable constant-energy trajectories. Using a SVR thermostat makes it possible to sample the NVT ensemble, with excellent agreement with the reference PET-C trajectories (Figure 17). The unstable MTS-16 trajectories, on the other hand, show large structural anomalies, that might be in part corrected by application of a suitably tuned colored-noise thermostat (Morrone et al., 2011). Given that the computational savings would be negligible, however, there is little reason to use $M$ greater than 8 when using non-conservative forces in the fast step.

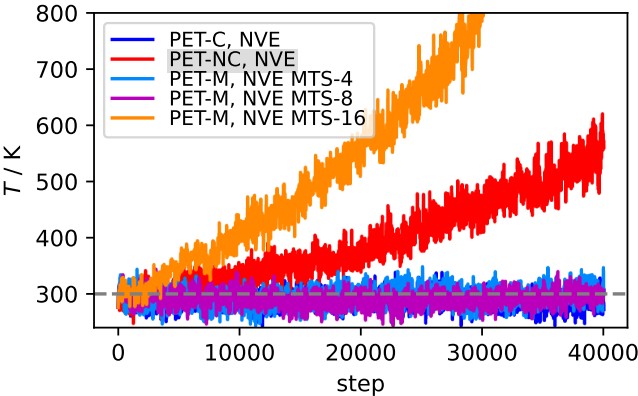

Figure 16. A comparison of kinetic temperature profiles for NVE MD trajectories of bulk water performed with the PET-M model and different orders of MTS integration. The step numbers refer to the "fast" evaluation.

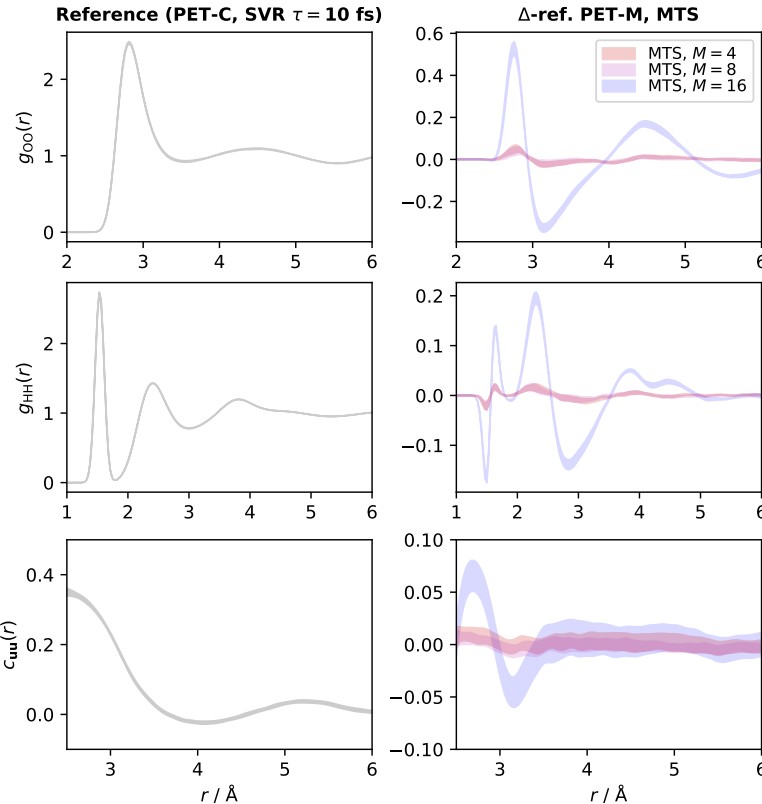

Figure 17. Structural correlations for simulations of room-temperature water using a PET-M model, using multiple time-step integrators to alternate the evaluation of non-conservative forces and (every $M$ steps) the conservative model. From top to bottom, the rows report the O-O correlation function, the H-H correlation function, and the orientational correlation function. All simulations use an efficient and gentle global SVR thermostat, with $\tau = 10$ fs coupling time. The left column shows the reference value (full PET-C MD) and the right column the differences with respect to this reference for MTS trajectories using different $M$ values. Shaded areas encompass two standard errors around the mean values.

We also investigate the behavior of MTS for the more generic PET potential trained on the OC20 dataset in Appendix H. We use the model after conservative fine-tuning, which is able to predict both conservative and non-conservative forces with a good accuracy. The simulations were run on a system of a single oxygen atom adsorbed on an alloy surface, and their kinetic energy profiles are shown in Figure 18. Also in this case, well-behaved molecular dynamics can be performed up to a high multiple time-step factor $M = 8$, a value that retains nearly all the speed-up afforded by non-conservative forces.

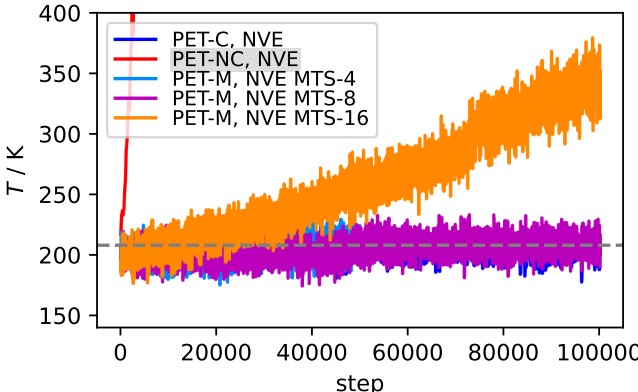

*Figure 18.* A comparison of kinetic temperature profiles for NVE MD trajectories of an oxygen atom adsorbed on an alloy surface, performed with the various heads of a model which was produced by "conservative fine-tuning" on a subset of the OC20 dataset. The step numbers refer to the "fast" evaluation.

## J. Software

Molecular dynamics simulations have been performed using the i-PI software (Litman et al., 2024), and geometry optimizations using ASE (Hjorth Larsen et al., 2017).

ORB, MACE-MP-0, Equiformer, SevenNet calculations were performed using the publicly available repositories, respectively `https://github.com/orbital-materials/orb-models/` (orb-models 0.4.0), `https://github.com/ACEsuit/mace` (mace-torch 0.3.8), `https://github.com/FAIR-Chem/fairchem` (fairchem-core 1.3.0), and `https://github.com/MDIL-SNU/SevenNet` (sevenn 0.10.1).

