# OpenReview forum: "The dark side of the forces: assessing non-conservative force models for atomistic machine learning"
_ICML.cc/2025/Conference — ICML 2025 oral_

### Official Review · Reviewer_J2Ug · 2025-02-21

**Overall Recommendation:** 5

**Summary:**

This paper investigates the implications of using machine learning models that predict non-conservative forces for atomistic simulations. Non-conservative models predict interatomic forces directly, rather than computing them as the derivative of a potential energy, which offers computational advantages but violates the fundamental principle of energy conservation. The authors compare conservative and non-conservative models and assess their performance in various simulation scenarios, including constant-energy (NVE) and constant-temperature (NVT) molecular dynamics (MD), as well as geometry optimization. They demonstrate that non-conservative forces lead to significant issues, such as ill-defined convergence in geometry optimization, energy drift in NVE MD, and artifacts in NVT MD, even with strong thermostatting. Further, the authors propose a hybrid approach that combines a conservative model with a non-conservative force predictor, which combines the benefits of direct force prediction (increased efficiency) with the stability and physical correctness of conservative forces.

## update after rebuttal

Please see comments below.

**Claims And Evidence:**

The central claim is that while non-conservative force models offer computational benefits, their use in atomistic simulations leads to fundamental problems due to the violation of energy conservation. This claim is well-supported by both theoretical arguments and empirical evidence.

**Essential References Not Discussed:**

The paper cites all essential references as far as I can judge.

**Experimental Designs Or Analyses:**

The experimental design and analyses are all sound and well-executed. It would have been interesting to also investigate less homogenous systems beyond bulk water (e.g., a solvated peptide) to investigate how general trends transfer between different systems (e.g., are non-conservative forces a bigger issue for heterogenous systems?), but this is a minor nitpick.

**Methods And Evaluation Criteria:**

The proposed methods and evaluation criteria are appropriate for the problem at hand and applied thoroughly.

**Other Comments Or Suggestions:**

I found two small typos in the supplementary information:
* Line 912 on p.17 ("without never" should be "without ever")
* Line 1028 on p.19 ("langevin" should be "Langevin")

**Other Strengths And Weaknesses:**

**Strengths**

+ The paper is very well-written and the arguments are presented logically and are easy to follow.
+ The paper addresses an important issue in the field of machine learning potentials, as non-conservative models are becoming increasingly popular and many practitioners seem unaware of the downsides of such models.
+ The authors provide a thorough investigation of the problem, covering both theoretical and practical aspects.
+ The authors propose a hybrid approach that combines the advantages of conservative (accurate and reliable) and non-conservative (fast) models.
+ The supplementary materials are very well constructed and provide many additional details to support the claims.

**Weaknesses:**

- The analysis is primarily focused on bulk water. While water is definitely a relevant system and I agree with its use as a paradigmatic example, it would be interesting to see how the findings generalize to other chemical systems. However, this is a very minor point (and the supplementary information also already contains some results for geometry optimizations of other systems).

**Questions For Authors:**

I have no important questions for the authors.

**Relation To Broader Scientific Literature:**

The paper clearly places the research in the context of current literature.

**Theoretical Claims:**

The theoretical claims are correct and well-explained. The derivation of the non-conservation metric (λ) is mathematically sound, and other statements in the paper are based on well-known facts and established theory.

---

> ### Author Rebuttal · Authors · 2025-03-31
>
> We thank the reviewer for the suggestion to investigate a solvated peptide, and have done so: we studied alanine dipeptide solvated in water, and additionally a benzene molecule adsorbed on a graphene surface, beyond the homogeneous cases of graphene, amorphous carbon and aluminium that we had already discussed in Appendix F. In all cases, the observation that non-conservative models like Equiformer and ORB display lack of energy conservation in NVE MD holds. The effect is comparable to or larger than that seen for homogeneous systems, and much larger than the drift observed for the more accurate single-purpose PET-NC model we trained for bulk water (https://imgur.com/a/CV3YmI3).
>
> As we argue with Reviewer NcLF, our choice of a comparatively simple system, for which very accurate models can be trained, is aimed at giving non-conservative predictions the most favorable treatment possible, and these additional tests confirm that the artifacts become even more severe when using general-purpose, universal models that are less accurate and entail more pronounced violation of energy conservation.
>
> We also thank the Reviewer for pointing out some typographical errors, which we have fixed.

---

> > ### Comment · Reviewer_J2Ug · 2025-04-03
> >
> > I thank the authors for also investigating a solvated peptide and benzene adsorbed on graphene. While not absolutely necessary, I believe these additional analyses make the paper even stronger.
> >
> > Thank you for this important contribution to the field!

---

### Official Review · Reviewer_NcLF · 2025-03-13

**Overall Recommendation:** 5

**Summary:**

The paper presents a systemmatic study of non-conservative force models for atomistic machine learning. Traditionally, forces are computed as the derivatives of potential energies to enforce symmetries and conservation laws. Several recent studies, however, directly predicted forces and learned the conservation laws implicitly during training to achieve a better trade-off between accuracy and computational efficiency. In this paper, the authors conducted microscopic simulations to investigate the non-conservative force fields. Experimental results show that these models suffer from several fundamental issues, from ill-defined convergence of geometry optimization to instability. In particular, energy conservation is harder to learn compared to rotational symmetry. The authors argue that the optimal way to use non-conserative force models is to integrate it with a conservative model, reducing the computation cost instead of completely eliminating the additional cost of backpropagation.

**Claims And Evidence:**

Most of the claims are supported by clear and convincing evidence. However, the following claim might be problematic to some extent:

1. The paper made claims on the limitations of non-conservative force fields, whereas the empirical study focuses primarily on bulk water as the benchmark system. While water is a good test case, it remains unclear whether the same effects would be observed in other materials.

**Essential References Not Discussed:**

All the relevant and essential references are discussed. There are two recent papers that also discuss the limitations of non-conservative force models, though they may not be perfectly within the scope of this paper.

[1] Loew, A. et al. Universal Machine Learning Interatomic Potentials are Ready for Phonons. Preprint at https://arxiv.org/abs/2412.16551v1.

[2] Fu, X. et al. Learning Smooth and Expressive Interatomic Potentials for Physical Property Prediction. Preprint at https://arxiv.org/abs/2502.12147.

**Experimental Designs Or Analyses:**

I checked the soundness and validity of the experimental design and analysis in the MD simulations and geometry optimization tests. The experiments are well-designed to test the claims made in the paper:

1. The paper compares multiple models, conservative or non-conservative, ensuring that findings are not tied to a single architecture.

1. NVE simulations effectively demonstrate the unphysical energy drift in non-conservative force models. NVT simulations evaluate whether thermostats can alleviate this issue. The use of both Langevin and SVR thermostats is appropriate for evaluating different correction methods.

1. The use of $\lambda$ to quantify deviations from the conserved energy is a novel and effective metric.

1. The manuscript provides the test results for different optimization methods (FIRE vs. L-BFGS) to demonstrate the impact of non-conservative forces on convergence.

**Methods And Evaluation Criteria:**

1. The authors conduct theoretical analysis and empirical validation on several benchmarks, including geometry optimization and molecular dynamics in NVE or NVT ensembles

1. The choice of bulk water as a benchmark dataset is appropriate, as it is widely studied and sensitive to force inaccuracies.

1. The authors use energy drift, Jacobian asymmetry, and kinetic temperature stability as the evaluation metrics which can directly capture the deficiencies of non-conservative models in simulations.

**Other Comments Or Suggestions:**

See Strengths and Weaknesses.

**Other Strengths And Weaknesses:**

Strengths:

1. The paper discusses the limitations of non-conservative force fields and points out that the best way to use non-conservative fore models is to integrate them into a hybrid model that follows physical constraints but also additionally supports direct, non-conservative,
force predictions. Therefore, instead of simply investigating the limitations, the paper provides a constructive pathway to mitigate the shortcomings of non-conservative force models, making them more applicable to real-world simulations.

Weaknesses:

1. The authors observe that thermostats cannot fully correct non-conservative behavior in NVT simulations, whereas only Langevin and SVR thermostats are tested. Exploring other thermostats (Nose-Hoover, adaptive thermostats, etc.) might provide a more complete picture.

**Questions For Authors:**

1. Why is the water dataset chosen for benchmarking instead of other systems? Is there any particular chemical insight that we can tell from benchmarking on the water dataset?

**Relation To Broader Scientific Literature:**

The paper reveals a deep insight into the limitations of non-conservative force fields which have been broadly studied in a lot of scientific literature for efficient force predictions. The paper points out that physics constraints should not be thoroughly removed, which provides a promising direction for the research in machine learning force fields.

**Theoretical Claims:**

I checked the mathematical derivation for the symmetry of Jacobians and it is correct.

---

> ### Author Rebuttal · Authors · 2025-03-31
>
> We thank the reviewer for pointing out additional relevant literature, which we have added to the manuscript. We also added the recent related work by Eissler et al. (arXiv:2503.01431), in which a non-conservative and non-equivariant transformer model is trained on a large dataset and used to study the degree of energy non-conservation in MD simulations.
>
> We chose water as a benchmark because it is an important system, ubiquitous in chemistry and biophysics, and because it is challenging due to the combination of high-frequency vibrations of the covalent O-H bond and the long-term dynamics of the hydrogen-bond network. Its relative compositional simplicity and homogeneity also means we can train a very accurate direct-force model, which allows us to show that non-conservative forces can produce unacceptable sampling artifacts even in this most favorable scenario.
>
> However, we agree that additional testing on more diverse molecular and materials systems is desirable. We had already included three examples (graphene, amorphous carbon, aluminum) in the appendices. Following the recommendation of this Reviewer (as well as Reviewer J2Ug), we performed additional tests on two heterogeneous systems: a benzene molecule adsorbed on a graphene surface, and alanine dipeptide solvated in water. We find that the non-conservative universal ORB and Equiformer models display severe lack of energy conservation in these cases, comparable to or worse than in the other systems we considered (https://imgur.com/a/CV3YmI3).
>
> Regarding the use of additional thermostats, we deliberately did not test Nosé-Hoover thermostats, which are non-ergodic in their original form and which, when using the “chain” extension [Martyna 1992] which improves (but does not fully fix [Patra 2014]) ergodicity by producing chaotic thermostat dynamics, amount essentially to a complicated strategy to generate poor-quality random numbers. We cannot think of a single case (besides perhaps the theoretical study of classical chaotic dynamics) in which N-H thermostats cannot be replaced by a stochastic thermostat. We agree with the Reviewer that it would be ideal to explore an “adaptive” thermostat to better substantiate our claim that thermostatting is not a valid fix for non-conservative dynamics. To that effect, we ran some tests using the “GLE-RESPA” [Morrone 2011] thermostats that use a bespoke generalized Langevin equation, tuned to stabilize noisy dynamics without disrupting the sampling of diffusive dynamics. Being local and adaptive, one could expect this approach to be preferable to a global (SVR) or plain Langevin thermostat for stabilizing direct force predictions. Our tests show that instead this GLE thermostat (which is less aggressive for slow degrees of freedom and hence does not slow down diffusion) is less effective at controlling the temperature drift induced by direct-prediction forces. The table below shows the atom-type resolved temperature for a simulation of bulk water at a target temperature of 300K, similar to table 2 in the manuscript. Results closer to 300K are better.
>
> | Model      | GLE (all atoms)         | GLE (H)           | GLE (O)           |
> |-----------------------|-------------------|-------------------|-------------------|
> | ORB (NC)              | 1181.5 ± 5.2      | 1099.2 ± 4.6      | 1362.5 ± 7.7      |
> | MACE (C)              | 303.1 ± 1.2       | 301.7 ± 1.1       | 306.0 ± 1.6       |
> | SevenNet (C)          | 304.1 ± 0.8       | 302.6 ± 1.3       | 307.3 ± 2.4       |
> | PET-NC (NC)           | 524.1 ± 4.6       | 495.5 ± 4.0       | 585.5 ± 5.9       |
> | PET-C (C)             | 301.8 ± 1.2       | 301.6 ± 1.2       | 302.2 ± 1.6       |
> | SOAP-BPNN-NC (NC)     | 1.6 × 10⁸         | 2.1 × 10⁸         | 5.9 × 10⁷         |
> | SOAP-BPNN-C (C)       | 301.3 ± 1.6       | 301.0 ± 1.4       | 301.9 ± 1.8       |
>
> This is consistent with our observation that non-conservative effects are more pronounced for long-range correlations and slow dynamics and reinforces our claim that thermostats can mitigate, but not solve, the artifacts induced by non-conservative forces. Finally, we would argue that the MTS strategy we propose provides a much better and more broadly applicable fix to the shortcomings of non-conservative forces compared to thermostats.
>
> [Martyna 1992] https://doi.org/10.1063/1.463940
>
> [Patra 2014] https://doi.org/10.1103/PhysRevE.90.043304
>
> [Morrone 2011] https://doi.org/10.1063/1.3518369

---

> > ### Comment · Reviewer_NcLF · 2025-04-07
> >
> > Thank the authors for the comprehensive rebuttal. The rebuttal along with the original manuscript has addressed most of my concerns. I think this is a very solid and insightful paper that will greatly contribute to the research in MLFF. With that, I'd like to raise my score from 4 to 5.

---

### Official Review · Reviewer_K2Jj · 2025-03-14

**Overall Recommendation:** 3

**Summary:**

This paper critically evaluates the implications of using non-conservative (NC) force models for learning machine learning interatomic potentials (MLIPs). This paper investigates the general accuracy of the NC models' forces, the effect of using NC force models on measured values of specific properties of interest in atomistic simulations, and geometric optimization.

While direct force prediction using NC force models provides a faster method for running atomistic simulations, the paper shows that even the best NC force models result in unstable simulations. The authors show that for a bulk water simulation, the NC forcefields result in a drift equivalent heating rate of about 7000 billion degrees per second. The authors also argue that some well-known correction procedures do not alleviate these issues. Using local thermostats to control the instability/run-away behaviors of NC force model-based simulation results can lead to slower convergence as it leads to a smaller diffusion coefficient, making it impossible to measure any time-dependent properties. Similarly, global thermostats can lead to the introduction of artificial artifacts as the non-conservative terms have different behaviors on different degrees of freedom. Through this experiment and other geometric optimization experiments, the authors convincingly show the unstable behavior introduced by using NC force models for atomistic simulations and geometric optimization.

Finally, the authors also provide a multi-step integration procedure for using hybrid models to effectively reap the benefits of the training and inference speed of NC force models while also enjoying the stability of conservation force models.

**Claims And Evidence:**

The authors claim that the NC force models do not implicitly learn force conservation and that the direct use of Non-conservative forces for atomistic simulation can lead to unstable simulations. The experimental setup and the evidence provided by the authors convincingly prove these claims.

The authors also claim that multi-step integrators with a hybrid multi-headed conservative and non-conservative force model can produce fast and stable simulations. While the authors show this claim to hold in one experiment, it is unclear if this still holds for more general systems and if the optimal M (shown to be 8 in the experiment considered) still leads to a non-trivial speed up in the simulations.

**Essential References Not Discussed:**

The authors have provided a detailed overview of the relevant related works.

**Experimental Designs Or Analyses:**

The experimental design and analysis is sound.

**Methods And Evaluation Criteria:**

The proposed metric of using lambda to understand the local deviations from the conservative fields and the measurement of the average kinetic energy and the velocity-velocity correlation function serve as appropriate and relevant measures for understanding the stability of the considered force models. Using the bulk water model provides a large and simple enough system to test the various problems about the stability of the NC force models.

**Other Comments Or Suggestions:**

N/A

**Other Strengths And Weaknesses:**

Strengths
- The paper brings up an essential issue in the field of MILPs.
- The proposed metrics and experiments provide a holistic picture of the problems associated with using Non-Conservative force models for simulations.
- The paper also highlights some problems with using current methods for making inference time corrections to improve the stability of atomistic simulations run using NC force models.

Weaknesses
- The paper does not provide ample evidence to support using hybrid models to fix some of the highlighted problems.

**Questions For Authors:**

Please refer to the previous sections.

**Relation To Broader Scientific Literature:**

As stated in the paper, many new works in MILPs are starting to look into direct force predictions without enforcing energy conservation. Generally, it has been seen that using physics-agnostic models trained at scale over enough data can lead to better models as the physical constraints can be learned from the data. This paper convincingly shows that ignoring energy conservation in learning Force models can lead to unstable atomistic simulations, and these problems cannot be easily fixed at inference time. This urges the field to either train better hybrid models using multi-head forces or develop better inference time methods to make the simulations more stable.

**Theoretical Claims:**

N/A

---

> ### Author Rebuttal · Authors · 2025-03-31
>
> We thank the Reviewer for their comments and questions. To provide more context, the use of multiple time-stepping to reduce the cost of a simulation while matching the accuracy of a “slow” potential is well-established in the molecular dynamics community. It was originally introduced to reduce the cost of long-range electrostatics (which is more computationally demanding, and harder to parallelize, than short-range components in a physically-motivated model) [Tuckerman 1992], and it has been used extensively together with different levels of electronic-structure theory [Marsalek 2016] and with machine-learning potentials [Rossi 2020]. It has also been recently applied to a very similar problem - namely correcting for symmetry breaking in unconstrained machine-learning models [Langer 2024]. In all these cases, and in that of non-conservative forces, MTS acts as a mitigation strategy, with smaller initial errors allowing for a larger stride M.
>
>
> Normalizing the cost of non-conservative MD to 1, and letting $F$ denote the slowdown for conservative force predictions, $2 \leq F \leq 3$, MTS reduces the theoretical cost of well-behaved molecular dynamics from $F$ to $1+(F-1)/M$. The evaluation of the conservative force is reduced by a factor $1/M$ and there is one common evaluation every $M$ steps. Note that in our current practical implementation, we do not yet re-use data for the common evaluation, and therefore incur a cost of $1+F/M$. We find that for the PET model employed in this work, $F\approx2$.
>
> In all our MTS experiments, the empirical timings match the theoretical ratios rather closely:
>
> | Type        | Relative speed (O on Sc-Co-Al alloy) | Relative speed (water) |
> | ----------- | --------------------- | -------------------- |
> | NC         |          1.0           |  1.0                    |
> | C            |          1.93        |  1.92                    |
> | MTS, M=4  |          1.50                  | 1.45                     |
> | MTS, M=8  |          1.19                  | 1.18                    |
> | MTS, M=16 |          1.05                  | 1.04                     |
>
> For $M=8$ and $F\approx2$, the simulation is only $\approx20$% slower than using exclusively non-conservative forces. Note that, even in the most unfavorable case (where only $M=2$ is stable), the theoretical computational overhead due to the use of conservative forces could be halved with an efficient implementation of MTS.
>
> Even though these general arguments suggest that MTS should be broadly applicable, we do agree with the Reviewer that further evidence that “hybrid” models are useful in general would be very valuable. To this end, we also consider the case of the diverse OC20 dataset (containing molecular fragments adsorbed on different catalytic surfaces) [Chanussot 2020].
>
> First, we performed a “conservative fine-tuning” exercise - reducing the training time by a factor of more than 3 while achieving the same final accuracy as the conservative model. This shows that the idea of pre-training with direct forces and then turning on backpropagation to fine-tune, therefore obtaining a hybrid model inexpensively, is far from specific to the water system. (Incidentally, this training method that we propose has already been picked up by the community to train state-of-the-art models, for example in [Fu 2025].)
>
> Subsequently, we use this model to perform MD on a representative structure (an oxygen atom adsorbed on an alloy surface). Even though the direct-force model shows a much worse temperature drift than observed in the experiments on water (consistent with the lower accuracy of the fit for this more broadly applicable potential) MTS still succeeds in stabilizing NVE trajectories up to M=8 (https://imgur.com/a/PY96n58). This provides a concrete example that MTS also works for less accurate, general-purpose direct-force models.
>
> To summarize, our work not only highlights the potential problems related to using non-conservative forces, but it also provides simple and effective solutions in the form of conservative fine-tuning (at training time) and MTS (at inference time), which allow practitioners to retain nearly all the advantages of using direct forces while avoiding their physical pitfalls.
>
> [Tuckerman 1992] https://doi.org/10.1063/1.463137
>
> [Marsalek 2016] https://doi.org/10.1063/1.4941093
>
> [Rossi 2020] https://doi.org/10.1021/acs.jctc.0c00362
>
> [Langer 2024] https://doi.org/10.1088/2632-2153/ad86a0
>
> [Chanussot 2020] https://doi.org/10.1021/acscatal.0c04525
>
> [Fu 2025] https://arxiv.org/abs/2502.12147

---

### Decision · Program_Chairs · 2025-05-01

**Decision:**

Accept (oral)

**Comment:**

The authors present a rigorous analysis of several nonconservative machine-learned force fields, showing how the deviations from energy conservation can lead to severely undesirable behavior. The reviewers all agreed the paper was high-quality, valuable work, an appreciated the inclusion of additional experiments beyond bulk water. The recommendation to accept was unanimous, with many reviewers strongly in favor, and I concur.